# Refining Marine Net Primary Production Estimates: Advanced Uncertainty Quantification through Probability Prediction Models

Jie Niu[1,2†], Mengyu Xie[3†], Yanqun Lu[3*], Liwei Sun[4], Na Liu[5], Han Qiu[6], Dongdong Liu[1,2], Chuanhao Wu[7], Pan Wu[1,2*]

[1]*College of Resources and Environmental Engineering, Guizhou University, Guiyang 550025, China*
[2]*Key Laboratory of Karst Georesources and Environment, Ministry of Education, Guiyang 550025, China*
[3] *Institute for Environmental and Climate Research, Jinan University, Guangzhou 510632, China*
[4]*Southern Marine Science and Engineering Guangdong Laboratory (Guangzhou), Guangzhou 511458, China*
[5]*College of Life Science and Technology, Jinan University, Guangzhou 510632, China*
[6]*Department of Sustainable Earth Systems Sciences, University of Texas, Dallas, Richardson, TX 75080, USA*
[7]*Yangtze Institute for Conservation and Development, Hohai University, Nanjing 210024, China*
*Corresponding author: Yanqun Lu, Pan Wu
E-mail address: Yanqunlv@163.com, pwu@gzu.edu.cn

[†] These authors contributed equally to this work and should be considered co-first authors.

## Abstract

In marine ecosystems, Net Primary Production (NPP) is important, not merely as a critical indicator of ecosystem health, but also as an essential component in the global carbon cycling process. Despite its significance, the accurate estimation of NPP is plagued by uncertainty stemming from multiple sources, including measurement challenges in the field, errors in satellite-based inversion methods, and inherent variability in ecosystem dynamics. This study focuses on the aquatic environs of Weizhou Island, located off the coast of Guangxi, China, and introduces an advanced probability prediction model aimed at improving NPP estimation accuracy while addressing its associated uncertainties. The dataset comprises eight distinct sets of monitoring data spanning from January 2007 to February 2018. NPP values were derived using three widely recognized estimation methods — VGPM, CAFE, and

CbPM — serving as model outputs for further analysis. The study evaluates two
probability prediction approaches: a Bayesian probability prediction model based on
empirical distribution and a deep learning-based probability prediction model. These
methods are employed to meticulously quantify the uncertainty of NPP. The results
highlight the effectiveness of probability prediction models in capturing the dynamic
trends and uncertainties of marine NPP. Notably, the neural network-based model
demonstrates superior accuracy and reliability compared to the Bayesian approach.
Furthermore, the models are applied to prognosticate NPP variations in specific marine
regions, efficaciously elucidating interannual trends. This research advances both the
methodological precision in quantifying NPP uncertainty and provides robust scientific
evidence for marine ecosystems management and environmental conservation.
**Keywords:** Net Primary Production; Bayesian Probability Prediction; Neural Network
Probability Prediction.

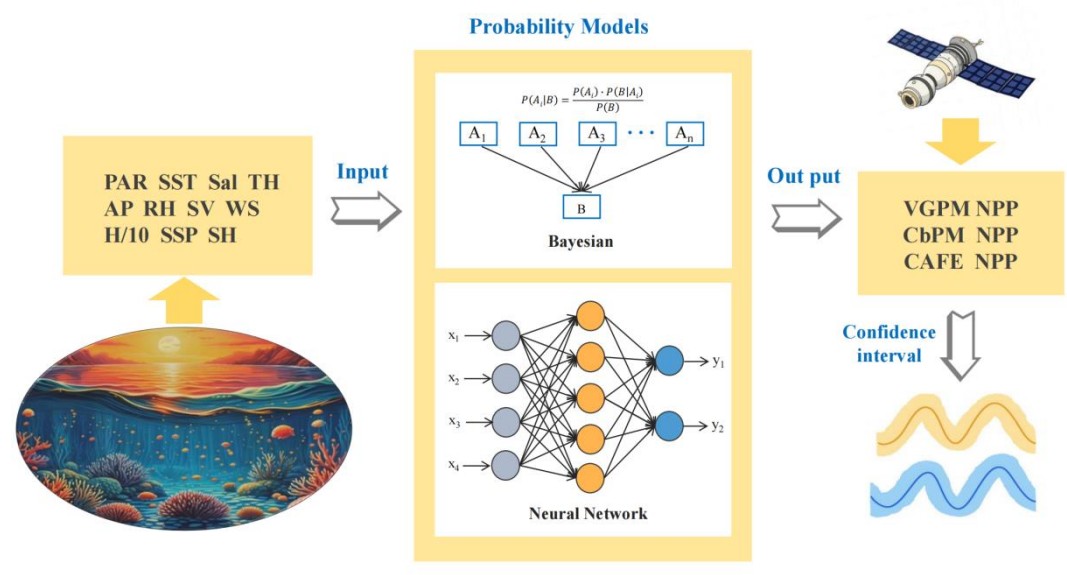

graphical abstract

## 1. Introduction

Net Primary Production (NPP) of phytoplankton, an essential indicator for
biological productivity, exerts a substantial influence on global carbon flux and the
dynamics of marine ecosystems (Yang et al., 2021; Silsbe et al., 2016). The precision
in estimating NPP is primary for environmental quality assessments (Falkowski et al.,
1998; Tan et al., 2005), effective fisheries resource management, and comprehending
the impacts of global climate change (Lee et al., 2015; Ding et al., 2016). Conventional
methods of NPP measurement, such as ship-based sampling and bottle incubations, are
beset with challenges like human errors and inadequacies in capturing spatial and
temporal dynamics. This underscores the necessity for more sophisticated and
comprehensive methods (Yang et al., 2021; Li et al., 2020).
The advent of ocean observation satellites and ocean color remote sensing
technology has catalyzed a paradigm shift in the estimation of large-scale marine
primary productivity (Yang et al., 2021; Westberry et al., 2008). These pioneering
technological advancements furnish novel insights into phytoplankton photosynthetic
production and its essential role in the carbon cycle, thereby broadening the
observational spectrum and establishing a robust foundation for predicting marine NPP.
Initial remote sensing endeavors to estimate NPP, employing satellite-based
chlorophyll-a (Chl-a) (Platt et al., 1991; Platt & Sathyendranath, 1988; Sathyendranath
et al., 1995), stemmed from the established correlation between chlorophyll and
photosynthesis (Ryther, 1956; Ryther & Yentsch, 1957). However, these efforts were
predominantly confined to local or regional applications. A subsequent investigation by
Campbell et al. (2002) delved into the accuracy of various satellite primary productivity
algorithms, unveiling that estimates from the most effective algorithm often diverged
from those derived from those obtained using the $^{14}$C isotope labeling method. Their
study also unearthed systematic biases in several algorithms, which could be alleviated
through re-parameterization. Satyendranath et al. (2020) emphasize the critical role of
accurately assigning parameters in primary production models as a key strategy for
reducing model uncertainties and enhancing the reliability of satellite-based primary
production estimates, particularly in the context of climate research.
Currently, the most widely utilized models for estimating NPP include the
Vertically Generalized Production Model (VGPM), the Carbon-based Productivity
Model (CbPM), and the Carbon, Absorption, and Fluorescence Euphotic-resolving
model (CAFE), have been proposed (Behrenfeld et al., 1997; Westberry et al., 2008;
Silsbe et al., 2016). Spanning various decades, these models address diverse facets of
ocean primary production and are readily accessible via satellite remote sensing data
platforms. As a result, they have been extensively applied and discussed in numerous
studies (Westberry et al., 2008; Pan et al., 2012; Dave et al., 2013; Li et al., 2020; Yang,
2021; Cael, 2021). Particularly, VGPM formulates a light-dependent, depth-integrated
model that classifies environmental factors influencing the vertical distribution and
optimal assimilation efficiency of primary production, leveraging $^{14}C$ productivity
measurement data (Behrenfeld et al., 1997). Conversely, CbPM was a depth-resolved
spectral NPP model designed for phytoplankton growth rates (Westberry et al., 2008).
Its foundational concept was originally articulated by Behrenfeld et al. (2005).
Distinguishing itself from Chl-based models, CbPM enables the differentiation of
physiological changes in biomass and Chl, thus offering a more nuanced depiction of
phytoplankton production. Notably, its strength lies in addressing issues related to light
and nutrient adaptation, thereby enhancing its capability in estimating fixed carbon
output at the ocean surface. Similarly, the CAFE model, introduced in 2016, presents
an adaptive framework that melds satellite ocean color analysis with essential
physiological and ecological attributes of phytoplankton (Silsbe et al., 2016). It
incorporates intrinsic optical properties into the model and calculates NPP by assessing
the product of energy absorption and the efficiency of converting absorbed energy into
carbon biomass, alongside computing growth rates. Nonetheless, these models
commonly generate a single value of NPP, overlooking the range estimation and the
inherent uncertainties in NPP estimation, stemming either from the model itself (BIPM
et al., 2009) or from the model input (Milutinovic & Bertino, 2011). This oversight is
critical, as suggested by Saba et al. (2011), since uncertainties in input variables, like
Chl-a, significantly impinge upon model performance and accuracy. In a recent
assessment, Westberry et al. (2023) examined the daily depth-integrated NPP rates over
2003–2018 for VGPM, CbPM, and CAFE, revealing that the mean NPP fields of CbPM
and CAFE, along with their associated frequency distributions, are distinctly divergent
from those of VGPM.
Transitioning from the constraints of traditional models, probabilistic forecasting,
in contrast to deterministic forecasting (Juban et al., 2007), generates a cumulative
distribution function or probability density function for the predicted object. This
methodology offers a more holistic understanding of likely outcomes (Gneiting &
Katzfuss, 2014; Schepen et al., 2018; Zhao et al., 2015). Significantly, this approach
has been successfully implemented in fields such as hydrology (Schepen et al., 2018;
Zhao et al., 2015; Schwanenberg et al., 2015) and power system management (Al-
Gabalawy et al., 2021). For instance, Schwanenberg et al. (2015) conducted analyses
using both deterministic and probabilistic forecasts. They concluded that deterministic
forecasts tend to overlook forecast uncertainty in short-term decisions, whereas
probabilistic forecasting offers numerous advantages: (i) it enables a longer forecast
horizon, facilitating earlier and more accurate predictions of major events; (ii) it
supports decision-making by incorporating forecast uncertainty into the analysis,
leading to more robust and adaptive outcomes; and (iii) it enhances the flexibility of
system operation through the integration of uncertainty-based methodologies.
The estimated values of NPP derived from the above three classical models
exhibit significant discrepancies, reflecting substantial uncertainties in these methods.
These inaccuracies can impede a comprehensive understanding of the role of oceans in
the global climate system, particularly in their capacity to act as carbon sinks and
regulators of atmospheric $CO_2$ levels. Consequently, quantifying and addressing these
uncertainties is primary to improving the reliability of NPP estimates and ensuring their
applicability in climate research and marine ecosystem management. Although
Bayesian models and probabilistic neural networks are established methods, their
application to the remote sensing of marine net primary productivity (NPP) represents
a novel approach. This study leverages these advanced probabilistic techniques to
address the unique challenges in estimating NPP from satellite data, providing a more

accurate and reliable quantification of uncertainties. We introduce probabilistic prediction models to meticulously quantify the uncertainty of NPP estimation, thereby enhancing our comprehension of NPP's significance in marine ecosystems. The research objectives of this paper are articulated as follows: (1) to thoroughly quantify the uncertainty of NPP estimation through the integration of probabilistic forecasting; (2) To evaluate and contrast the efficacy of neural network-based probabilistic forecasting with empirical distribution-based Bayesian probabilistic forecasting in capturing NPP uncertainty; and (3) To implement probabilistic forecasting of the uncertainty of the NPP in the study area during 2007–2018 and to explore its temporal characteristics. Our study offers innovative perspectives and methodologies for addressing the uncertainty associated with NPP. The organization of this paper is as follows: Section 2 outlines the study area and data sources; Section 3 elaborates on the methodology and presents metrics for evaluating forecasting performance; Section 4 discusses the results; and Section 5 presents the conclusions.

## 2. Data and Methods

### 2.1. Study Area and Data Sources

The research locale for this study is situated in the aquatic environs of Weizhou Island, nestled within the Gulf of Tonkin, Guangxi Province, southern China (Fig. 1). The proportion of excellent water quality in Guangxi's near-shore waters reaches more than 90% all year round, and the quality of the marine ecological environment has remained at the forefront of the country for 12 consecutive years, which is the only stable habitat and feeding ground for large cetaceans known in China's near-shore waters at present. Weizhou Island is the youngest volcanic island in China geologically, with more than 95% of its strata comprising volcanic rocks. Its landscape features include formations resulting from sea erosion, marine sediment accumulation, and dissolved rocks. Weizhou Island, located in the southern subtropical monsoon zone, experiences a pleasant climate with abundant heat and precipitation throughout the year. The average annual temperature is 23℃, and the average winter temperature is 16.3℃.

The unique climatic conditions and island landscape make it a popular tourist destination. The waters of Weizhou Island are the habitat of many rare marine organisms, and the protection and research of its marine ecosystem are of great significance to maintaining marine biodiversity.

The dataset of this study encompasses eight distinct sets of monitoring data spanning from January 2007 to February 2018, amassing a total of 4077 days. These data were procured from the Weizhou Marine Environmental Monitoring Station (21.0017°N, 109.0117°E) and encompass a spectrum of variables: sea surface temperature (SST), salinity (Sal), tide height (TH), air pressure (AP), relative humidity (RH), sea visibility (SV), wind speed (WS), and 1/10th significant wave height (H/10). Additionally, photosynthetically active radiation (PAR) was retrieved from NASA's Ocean Color portal (https://oceancolor.gsfc.nasa.gov/), sea surface precipitation (SSP) was sourced from Nasa Earth Observation Data (https://www.earthdata.nasa.gov/), and sunshine hours (SH) was sourced from the China Meteorological Administration (https://data.cma.cn/). These data were aggregated to constitute a comprehensive dataset encompassing eleven variables, serving as the input features for the models. Phytoplankton, the primary source of NPP, is directly influenced by variables such as SST, Par, and SH, which are critical to its photosynthetic processes. Additionally, other variables have significant indirect effects on phytoplankton growth. Sal, for example, influences the community structure of phytoplankton (Braarud et al., 1951). Variables such as TH, H/10, and WS indirectly affect phytoplankton dynamics by modulating water column mixing and the vertical distribution of nutrients. AP, RH and SV also indirectly impacts phytoplankton photosynthetic activity by altering environmental conditions. For the analysis of three NPP algorithms—namely, VGPM, CbPM, and CAFE—we utilized their output datasets, which were obtained at an eight-day temporal resolution from the Ocean Productivity website (http://orca.science.oregonstate.edu/1080.by.2160.monthly.hdf.vgpm.m.chl.m.sst.php, http://orca.science.oregonstate.edu/1080.by.2160.monthly.hdf.cbpm2.m.php, http://orca.science.oregonstate.edu/1080.by.2160.monthly.hdf.cafe.m.php). These

datasets represent the modeled NPP estimates produced by each algorithm over a
cumulative duration of 514 days. The specific datasets utilized for this study are
itemized in Table 1.
Due to factors such as equipment malfunctions and adverse weather conditions,
some data for the eleven variables were incomplete. To gain a deeper understanding of
the data structure and address these gaps, we conducted an analysis of the missing data
and identified five variables with missing entries (Table 2): SV, H/10, SSP, PAR, and
SH. These missing data points are primarily due to random occurrences such as satellite
equipment malfunctions and severe weather conditions, which disrupt data acquisition.
Since these events are sporadic and not tied to any specific frequency, only the total
number of missing values has been recorded. Subsequently, we visualized these five
variables in a chronological sequence, with the findings depicted in Fig. 2. Distinct
from daylength, which is computable based on location and date, SH indeed refers to
the daily measured duration of sunlight reaching the Earth's surface. The variability and
instances of zero values observed in Fig. 2 (bottom panel) and mentioned in Table 2
reflect real-world fluctuations due to weather conditions—on overcast or rainy days,
actual sunshine hours recorded can indeed drop to zero. These data are collected on a
daily basis, hence the seemingly sporadic pattern rather than a smooth temporal
variation expected of constant daylength calculations. The analysis revealed a marked
periodicity in these variables, prompting us to employ time series interpolation as our
method of choice for data imputation. The efficacy of this approach is evidenced in
Table 3, which presents the statistical indicators of the data both pre- and post-
interpolation. Notably, while the post-interpolation data retains a close resemblance to
the original data in terms of statistical indicators, it is important to acknowledge that
interpolated data are not independent observations. The validity of the interpolation
method, therefore, depends on the specific application and context. In this study,
interpolation was used to address missing variables, and we ensure that the statistical
properties of the original data were preserved to the greatest extent possible. This
approach allows us to maintain the integrity of our analyses while recognizing the
inherent limitations of using interpolated data.
VGPM, CbPM, and CAFE rely on similar input variables, derived from satellite
observations and environmental measurements. VGPM uses inputs such as SST,
chlorophyll concentration (Chl), and PAR to estimate NPP, leveraging optimal
assimilation efficiency in its parameterization (Behrenfeld et al., 1997). CbPM focuses
on phytoplankton carbon biomass, incorporating backscattering coefficients along with
Chl. CAFE integrates additional inputs, including atmospheric pressure (AP), solar heat
(SH), and wind speed (WS), to parameterize light and nutrient availability critical for
phytoplankton growth.
To evaluate the long-term trends in Net Primary Production (NPP), we applied a
low-pass filter to the three NPP products (VGPM, CbPM, and CAFE) (Fig. 3). This
filtering process removes high-frequency variations, such as noise and short-term
fluctuations, while retaining the underlying long-term patterns. It became evident that
each exhibits a distinct seasonal periodicity, with the fluctuation ranges remaining
stable over time yet the magnitude and timing of them varing significantly among the
three NPPs. Specifically, VGPM are the smallest, followed by CAFE, while CbPM
have the largest values. This periodicity indicates that changes in NPP are not random
but follow predictable laws and reflects the well-established seasonal patterns in marine
primary production, associated with seasonal variations in environmental factors such
as light availability, temperature, and nutrient. Such periodic trends are expected in
regions around 21 degrees north, including the waters near Weizhou Island, due to the
interplay of monsoonal influences and seasonal shifts in oceanographic conditions.
While all three NPPs capture these periodic patterns, their representation of the
magnitude and timing of peaks differs. The distinct ways in which VGPM, CbPM, and
CAFE capture these patterns provide valuable insights into their respective model
designs and parameterizations.
To elucidate the correlation between these NPP products and our dataset, we
generated Pearson correlation plots (Fig. 4). The results revealed that the variables with

the highest correlations differed among the three NPP values. Notably, VGPM showed the strongest correlation with SST, reflecting its dependence on sea surface temperature in its parameterization. Both CAFE and CbPM showed strong correlation with AP, albeit in opposing directions—CAFE displayed a positive correlation, while CbPM NPP exhibited a negative one. Changes in AP affect atmospheric stability, cloudiness, and precipitation, indirectly altering light conditions in the ocean and subsequently affecting phytoplankton photosynthesis. Lower AP often corresponds to unstable atmospheric conditions and increased cloud cover, which may inhibit photosynthesis activity by reducing light penetration. Additionally, phytoplankton dynamics modeled in CbPM may respond differently to such changes compared to CAFE, potentially due to the distinct assumptions and parameterization used in each model. In summary, among the three models, VGPM possesses the most significant correlation with the variables, followed by CAFE, and lastly CbPM.

## 2.2. Methods

### 2.2.1. Bayesian Probability Prediction

Bayesian models can adeptly quantify the uncertainty in the distribution of predicted outcomes. The Bayesian approach is particularly advantageous in scenarios with limited training data or when potential invisibility in training data cannot be discounted in practical applications (Perfors et al, 2011; Kaplan D, 2021; Zou et al, 2024). The Bayesian formula is represented as:

$$P(\theta|D) = \frac{P(D|\theta) \cdot P(\theta)}{P(D)} \tag{1}$$

where $P(\theta|D)$ denotes the posterior probability, $P(D|\theta)$ the likelihood probability, $P(\theta)$ the prior probability, and $P(D)$ the marginal probability for normalization.

When a training dataset D is available, the probability distribution $P(\theta|D)$ of $\theta$ is computable using the aforementioned Bayesian formula (Dürr et al, 2020). To deduce $P(\theta|D)$, it is imperative to ascertain the likelihood probability $P(D|\theta)$ of the observed

data under the model parameter θ. P(D|θ) can also be interpreted as the probability of
obtaining the training dataset D given parameter θ. Additionally, knowledge of the prior
probability P(θ) and the evidence P(D) is essential. Given that the training dataset D is
fixed, P(D) remains constant. Consequently, the posterior distribution is proportional to
the likelihood probability multiplied by the prior distribution, i.e., $P(\theta|D) \propto P(D|\theta) \cdot$
$P(\theta)$, in accordance with Bayes' Law.
In this study, the Bayesian approach is employed to calculate the posterior
distributions of the parameters considering the prior information and the input data.
Subsequent predictions are made using the posterior distributions, yielding a
probability distribution for each predicted value. Ultimately, the model's ability to
estimate the uncertainty in the NPP is illustrated by plotting the prediction ranges for
different targets and comparing them to actual observations.
2.2.2. Neural Network Probabilistic Prediction Model Based on TFP
TensorFlow Probability (TFP) represents a sophisticated library of statistical
algorithms, devised atop the TensorFlow Python API. Its primary objective is to
streamline the integration of probabilistic models with deep learning frameworks. TFP
offers a comprehensive suite of tools, enabling the construction of probabilistic models
adept at estimating uncertainty. Aiming to thoroughly assess the predictive efficacy of
the three NPP products, we employed a neural network model grounded in the TFP
framework, capitalizing on its versatility and potent expressive capabilities for
probabilistic prediction in marine ecosystems.
The architecture of this neural network model incorporates multiple hidden layers,
each implementing a nonlinear transformation via an activation function. Such a
configuration enables the model to automatically extract higher-order features and
intricate patterns from the data. Our selection of TFP as the implementation medium
allows us to model the neural network's output by integrating probability distributions,
thus addressing the model's uncertainty regarding predictions and yielding more
exhaustive insights. Specifically, our neural network model utilizes a distribution layer
in the output stage, producing a probabilistic distribution concerning the target variable,
as opposed to a mere deterministic point prediction. This probabilistic output facilitates
the quantification of the model's confidence level for each prediction, extending beyond
mere point estimates.

The integration of Bayesian models and probabilistic neural networks in our

approach addresses key challenges in the remote sensing of NPP. These challenges
include handling the variability and uncertainty inherent in satellite-derived data and
environmental factors, thus improving the robustness of NPP estimates. In this study,
the input variables for the models are the 11 environmental variables mentioned in
Section 2.1, and the outputs are VGPM, CbPM, and CAFE. These inputs overlap
substantially with those used in VGPM, CbPM, and CAFE, demonstrating that the NN
and Bayesian models do not require additional or more complex inputs. The selection
of input data was not limited to variables directly related to phytoplankton
photosynthesis, such as SST, PAR, and SH. Instead, it also included a wide range of
environmental variables that could influence phytoplankton growth, such as TH, WS,
and AP, which are physical dynamics and meteorological characteristics. Since
phytoplankton are the primary source of NPP, environmental factors affecting
phytoplankton growth also indirectly impact NPP. These emphasize the variability in
how different NPP models capture environmental interactions. Importantly, the Pearson
correlation analysis (Fig. 4) highlights the most relevant variables for prediction,
enabling the NN and Bayesian models to focus on key inputs and filter out less
influential variables.

The dataset spans 4,077 days, but due to the 8-day time interval of the downloaded

NPP products, only 514 complete datasets are available for model training and
performance evaluation. Given the limited amount of data, 80% of the 514 sets are used
for model training and parameter tuning, while the remaining 20% are used for
performance evaluation. In the neural network probabilistic prediction model, there are

six layers, with two output nodes used to estimate the mean and standard deviation. The Gaussian distribution is employed in the distribution layer, and the loss function is the negative log-likelihood loss function. The detailed parameters of the neural network are presented in Table 4.

## 2.3. Model Evaluation

Prior to model evaluation, we normalized the NPP satellite data. This step is critical to improving model performance because it removes the potential effects of different data scales, allowing the model to consider each data point more fairly. Normalization ensures that the distribution range of NPP data has the same weight during model training, thus improving the model's ability to capture the inherent patterns and features of the data. In addition, normalization helps reduce the noise and bias introduced by data scale differences, further enhancing the stability and predictive accuracy of the model.

Before training the model, we divided the dataset reasonably. Specifically, we divided the dataset into 80% training set and 20% testing set. This division aims to ensure that the model can fully learn the features and patterns of the data during the training process, while retaining enough independent data for testing the predictive ability of the model. This way of dividing the dataset helps us to evaluate the performance of the model more accurately and avoid problems such as overfitting.

In this study, our models provide probabilistic predictions, generating a probability distribution for each time point rather than a single point estimate. To facilitate visualization and interpretation, the curves presented in some figures represent the mean values derived from these predictive distributions. These mean curves summarize the central tendency of the model outputs while inherently accounting for the uncertainty associated with the predictions.

### 2.3.1. CRPS

Continuous Ranked Probability Score (CRPS) is a sophisticated statistical metric employed to evaluate the efficacy of forecasting models. Initially introduced in the 1970s (Matheson & Winkler, 1976), CRPS is widely utilized in areas such as weather forecasting (Zamo et al., 2018). It quantifies the divergence between the predicted probability distribution and the actual observations (Hersbach, 2000). Ideally suited for scenarios where the target variable is continuous and the model predicts its distribution (Pic et al., 2023), CRPS equates to the mean absolute error (MAE) in deterministic forecasting (Zhao et al., 2015).

In probabilistic forecasting, the focus extends beyond mere point estimates to encompass the shape and dispersion of the probability distribution. Hence, traditional scoring functions prove inadequate, as aggregating the predicted distributions into their mean or median neglects critical information about the dispersion and shape. CRPS, by embracing the entire probability distribution, emerges as an invaluable tool in assessing model uncertainty. CRPS is calculated as follows:

1. For each sample (individual data points in the dataset, each representing a specific combination of environmental conditions and corresponding NPP estimates), calculate the discrepancy between the cumulative distribution function (CDF) of the predicted and observed values.

2. Aggregate the variances for all samples and divide by the number of samples to obtain the average variance.

$$CRPS_{individual}(F,x) = \int_{-\infty}^{+\infty}[(F(y) - H(y - x)]^2 dy \tag{2}$$

$$CRPS = \frac{1}{n}\sum_{i=1}^{n} CRPS_{individual}(F_i, x_i) \tag{3}$$

where $F(y)$ denotes the CDF of the predicted value, $y$ the predicted value, $x$ the observed value, and $H(y\text{-}x)$ the Heaviside function which is 0 when $y<x$ and 1 otherwise. $n$ indicates the total number of samples, and $CRPS_{individual}(F_i, x_i)$ the CRPS value for the

*i-th* sample.
A smaller CRPS value signifies a closer alignment of the model's probability
distribution with actual observation, integrating insights on both the shape and location
of the distribution and demonstrating sensitivity to outliers. Unlike other metrics such
as Root Mean Square Error (RMSE) or Mean Absolute Error (MAE), CRPS offers a
more holistic evaluation of a probability distribution's predictive capacity by
considering the full distribution shape. For Bayesian and neural network models,
comparing CRPS values facilitates an understanding of their proficiency in fitting the
entire probability distribution.
2.3.2. CDF
The Cumulative Distribution Function (CDF), also known as the distribution
function, is the integral of probability density function (PDF). It provides a complete
description of the probability distribution of a real-valued random variable $X$. The CDF
is defined as the probability $P$ that a random variable $X$ is less than or equal to a given
value $x$, expressed as:
$$F(x) = P(X \leq x) \tag{4}$$

To evaluate the predictive performance of the model, we computed the empirical
CDF of the input data and compared it with the average predictive CDF generated by the
model. This comparison provides a graphical representation of the model's predictive
accuracy. A higher degree of overlap between the empirical and predictive CDF curves
indicate a greater similarity between the two distributions, thereby reflecting superior
model predictions.
2.3.3. RMSD
Root Mean Squared Deviation (RMSD) is a widely recognized evaluation metric
in regression analyses, primarily employed to quantify the discrepancy between a

model's predicted values and the actual observed values. Characterized by its intuitive nature and simplicity in computation, RMSD is particularly beneficial in scenarios where emphasis is placed on the magnitude of difference between predicted and actual values, irrespective of the difference's direction.

$$RMSD = \sqrt{\frac{1}{n}\sum_{i=1}^{n}(y_i - x_i)^2} \qquad (5)$$

where $n$ denotes the number of samples, $y_i$ represents the predicted value of the *i-th* sample, and $x_i$ symbolizes the actual value of the *i-th* sample.

A lower RMSD value is indicative of superior model performance, signaling a smaller variance between the model's predictions and the observed values. Nevertheless, it is important to note that RMSD exhibits sensitivity to outliers, as it constitutes the mean of the squared differences. Incorporating RMSD alongside CRPS in our analysis enables a more comprehensive evaluation of both the overall accuracy and uncertainty inherent in the predictions.

### 2.3.4. MAPD

Mean Absolute Percentage Deviation (MAPD) is a frequently utilized percentage error metric in regression problems. It expresses the prediction error as a percentage, offering an insightful perspective into the relative error between predicted results and true values in predictive model evaluations.

$$MAPD = \frac{1}{n}\sum_{i=1}^{n}\left|\frac{x_i - y_i}{x_i}\right| \times 100\% \qquad (6)$$

where $n$ signifies the number of samples, $y_i$ the predicted value of the *i-th* sample, and $x_i$ the actual value of the *i-th* sample.

A lower MAPD value is desirable, indicating a reduced relative error of the model. However, a cautionary note: MAPD may prove unreliable in instances where the predicted value approaches zero, as a zero denominator results in infinity. Therefore,

careful consideration is warranted when employing MAPD, particularly in scenarios where relative accuracy is primary.

In the context of comparing Bayesian probabilistic prediction models with neural network probabilistic prediction models, the synergistic application of these three metrics—CRPS, RMSD, and MAPD—affords a multifaceted assessment of the models. This triad of metrics enhances our understanding of the importance of relative error alongside the accuracy of point estimates and the fit of probability distributions.

## 3. Results and Discussion

### 3.1. Comparative Analysis of Prediction Efficacy Between Two Models

We utilized VGPM, CbPM, and CAFE as prediction targets to scrutinize the predictive effectiveness of both the neural network-based probabilistic prediction model and the empirical distribution-based Bayesian probabilistic prediction model. Fig. 5 presents a comparison of CRPS, RMSD, and MAPD values for both NN and Bayes models using three NPPs as prediction targets across training and test datasets. Notably, CRPS provides a holistic evaluation of prediction accuracy and reliability. All the metrics are calculated using normalized data for better comparison. Lower values are indicative of enhanced model performance. Fig. 5(a)-(c) and (d)-(f) respectively depict the CRPS, RMSD, and MAPD of the NN model and Bayes model when using the three NPP values as prediction targets. The color blue represents the training set, while red represents the test set. It can be observed from Fig.5 (a) and (d) that the CRPS values of both the NN model and Bayes model are similar. When VGPM is used as a prediction target, the performance of the models is closest between the training set and test set, followed by CbPM. However, CAFE has the lowest CRPS value among all three models, with its test set slightly larger than that of its training set. The lower CRPS value for the CAFE NPP, compared to VGPM and CbPM, may stem from the fact that its probability distribution aligns more closely with the prediction of models in terms of both shape and central tendency, since CRPS evaluates the full probability

distribution, incorporating factors such as skewness and kurtosis in addition to variance. In the case of CAFE, the probabilistic structure of its predictions may exhibit better congruence with the observed cumulative distribution function (CDF) (Section 3.2.2), particularly in regions with higher data density. This enhanced alignment could compensate for its slightly larger variance compared to CbPM, thereby resulting in a lower CRPS value. Additionally, the design and parameterization of the CAFE model may inherently emphasize features that lead to improved probabilistic predictions, which warrants further investigation.

In terms of RMSD metrics (Fig. 5 (b) and (e)), when VGPM is used as a prediction target, its index value is significantly higher compared to others; however, its performance between training set and test set remains close. When CbPM is used as a prediction target, Bayes model outperforms NN model but exhibits a larger difference between training set and test set compared to NN model.

The neural network and Bayesian models developed in this study were trained using outputs from the VGPM, CbPM, and CAFE models. While this approach allowed us to evaluate the uncertainty in emulating these base models, it also means that our models inherit their underlying biases and errors. As such, the uncertainty estimates reported here reflect the uncertainty in emulating these specific outputs and do not represent the true uncertainty of NPP estimation. Furthermore, as Fig. 3 demonstrates, the outputs of VGPM, CbPM, and CAFE differ significantly, underscoring the need for ground truth data to validate these models. Among these, CAFE NPP is often considered more accurate based on prior studies, but further validation with observational data is necessary to confirm this assumption.

On using CAFE as a prediction target, both models show more consistent performance. The values of these indicators are relatively close in all aspects at around 0.2. Regarding MAPD metrics (Fig.5 (c) and (f)), clear differences among the three NPP models can be seen where CAFE has obviously lower index value compared to CbPM and VGPM. In addition, for NN model's MAPD index value for CAFE is lower

than that for Bayes model. However there exists significant difference between its training set and test set.

Overall evaluation indicates that under both models' assessment criteria, CAFE demonstrates superior accuracy in predicting effects compared to VGPM and CbPM. VGPM shows greater instability with inferiority in its training process over testing process (Fig.5 (d), (e), (f)), which may be attributed to overfitting. However, there is a more noticeable difference in the performance of CbPM in the two models. The CRPS value and RMSD value in the Bayes model are significantly lower than those in the NN model (Bayes is less than 0.2, while NN is more than 0.2).

Therefore, among the three NPP datasets (VGPM, CbPM, and CAFE), the CAFE was selected as the primary prediction target for subsequent analysis. This decision was motivated by two factors: (1) prior research indicating that CAFE provides relatively accurate estimates of NPP in marine ecosystems with characteristics similar to the Weizhou Island area, due to its advanced parameterization of phytoplankton dynamics, and (2) the demonstrated ability of both probabilistic prediction models (NN and Bayesian) to emulate CAFE output with high accuracy and reliability. While this does not imply that CAFE perfectly represents true NPP, its suitability for capturing patterns in the study area supports its use as the prediction target in this work.

## 3.2. Quantify the Uncertainty of CAFE

When quantifying uncertainty in the CAFE, we need to focus on the uncertainty factors that exist in the input variables in addition to the uncertainty that may arise during model training. These uncertainty factors include measurement errors and temporal variability, among others. Measurement errors usually originate from the accuracy limitations of the instruments, the complexity of the observation environment, or the instability of human operations. These errors not only affect the accuracy of the input variables to varying degrees, but also propagate through the model and thus affect the accuracy of the prediction results. The temporal variability, on the other hand,

reflects the dynamic changes of marine environmental parameters, such as seasonal temperature changes, cyclic fluctuations of tides, etc., which also affect the NPP prediction results. Consequently, quantifying these uncertainties is particularly important in conducting CAFE predictions.

3.2.1. Comparative Analysis of Confidence Interval Widths

Fig. 6 illustrates the comparison between the forecast mean of the NN model and Bayes model, and the CAFE value when CAFE is utilized as the prediction target. In the figure, the triangular icons represent 514 sets of the forecast average, while the gray and blue represent the 95% and 75% confidence intervals, respectively. Overall, both models exhibit relatively wide confidence intervals for their predicted results, possibly due to the large range of changes in CAFE. The models may face greater challenges in capturing this wide range of changes, resulting in increased uncertainty.

When CAFE is less than 450 mg C m$^{-2}$ d$^{-1}$, both models tend to overestimate the actual NPP value. This phenomenon becomes more pronounced when CAFE is less than 350 mg C m$^{-2}$ d$^{-1}$. In contrast, a certain linear relationship between true value and predicted mean value emerges within a range of 450-600 mg C m$^{-2}$ d$^{-1}$. Most of the predicted mean values are distributed around the 1:1 line in this range, indicating higher accuracy by these models. However, when CAFE exceeds 600 mg C m$^{-2}$ d$^{-1}$, it is observed that both models tend to underestimate actual NPP values. This phenomenon may be attributed to an imbalance in sample data distribution within different intervals of CAFE. The majority of data points are concentrated in a narrow range (350-600 mg C m$^{-2}$ d$^{-1}$), while data points in other intervals are scarce. This inadequacy makes it difficult for model training to capture its distribution law accurately and leads to increased prediction uncertainty within these ranges.

Compared with the two models, the predicted value of NN model is more concentrated around the 1:1 line, while the predicted value of Bayes model is relatively dispersed and the confidence interval is wider. The smaller the confidence interval width, the higher the accuracy of model prediction. It manifests that the NN probabilistic prediction model is more accurate in predicting CAFE than the Bayes

probabilistic prediction model, and the uncertainty of its prediction results is lower. The prediction mean obtained by the NN probabilistic prediction model is closer to the 1:1 line, which usually means that the deviation between the predicted value of the model and the actual observed value is small, that is, the prediction accuracy of the model is higher. The differences in the performance of the two models may stem from their different strategies for dealing with uncertainty and data fitting. Neural network models typically capture the nonlinear relationships of data through a large number of parameters and complex network structures, so they may be able to fit the data distribution more accurately in some cases. Bayes model deals with uncertainty by introducing prior knowledge and a posteriori inference, but its performance may be limited under some complex data distributions.

To further elucidate the models' effectiveness in probabilistic prediction of CAFE, Fig. 7 visualizes the time series model predictions with a 95% confidence interval uncertainty range. The figure shows that almost all CAFE values fall within the 95% confidence interval of the mean of the predicted values. It can be clearly seen that the predicted distribution of the NN model is much smaller than that of the Bayes model, which is consistent with the results shown in Fig. 6. The NPP is clearly periodic in time, and both models are able to align their predictions on the test set with the periodicity of the training set. In particular, the scatter in the NN model is more centrally distributed around the red line, while the scatter in the Bayes model is more discrete from the red line, which further suggests that the NN model has a more accurate estimate in predicting the CAFE.

Overall, the trends in the predicted means of the two models are consistent with the trends in the majority of CAFE values, which further validates the accuracy of the two methods in capturing the process of CAFE changes. This consistency not only indicates that the models can accurately reflect the long-term trends of CAFE changes, but also capture short-term fluctuations and outliers. This is of great significance for ecosystem monitoring and prediction, and helps to better understand the dynamics of the ecosystem and take appropriate management and conservation measures. However,

in terms of confidence interval width, the width of the 95% confidence interval in the results of the Bayesian probabilistic prediction model is larger than that of the neural network probabilistic prediction model, indicating that the Bayesian probabilistic prediction model is not as sharp as the neural network probabilistic prediction model, which is more locally sensitive and able to respond to the changes in data more quickly.

Although the neural network probabilistic prediction model shows an advantage in terms of sharpness and local sensitivity, this does not mean that it is superior to the Bayesian model in all cases. In fact, Bayesian models are more robust and explanatory by introducing prior knowledge and posterior inferences to deal with uncertainty. Therefore, when choosing a predictive model, trade-offs need to be made based on specific application scenarios and data characteristics.

3.2.2. Comparative Analysis of CDF

Fig. 8 depicts the overall predictive distribution versus the empirical distribution of the CAFE input data. Concurrently, Fig. 9 methodically quantifies the disparity between the average predictive CDF and the empirical CDF of the input data. Optimally, the divergence between these two CDFs should be minimal, manifested as extensive overlap between the yellow and blue curves in Fig. 8, and the blue curve in Fig. 9 approaching zero. Fig. 8 demonstrates the CDF curves of the predicted mean values after the normalization process and the CDF curves of the CAFE. The CDF plots of the normalized data can reflect the statistical distribution of the datasets, especially when the different datasets have different magnitudes or scales, and the normalization can eliminate these differences, which makes the comparisons and analyses between the different datasets more accurate and intuitive. Fig. 9 specifically quantifies the difference between the two CDF curves in Fig. 8 at each point, which is accomplished by calculating the difference between the y-values of the two CDF curves at the same x-value.

While the cumulative distribution function (CDF) curves in Fig. 8 show apparent differences between the test and train datasets for CAFE, these differences can primarily be attributed to the smaller size of the test dataset relative to the training

dataset. Such size discrepancies can cause the CDF curves to appear visually different, even when the underlying data distributions are similar. Moreover, as shown in Fig. 7, the patterns for simulating the training set and predicting the test set are consistent for both the NN and Bayesian models. This consistency indicates that the models generalize well to the test data, capturing its key characteristics despite the visual differences in the CDF curves. Therefore, the observed discrepancy in the CDF curves does not imply poor representation of the test data by the training data. For the NN probabilistic prediction model, when the CAFE values are lower, the two CDF curves on the training set and the test set move gently and almost overlap, with the difference close to 0, which indicates that the model can predict the actual data distribution well within the range of small values of CAFE. As CAFE increases, the difference between the predicted and true CDF curves grows larger, with the predicted mean CDF on the training set generally lying below the CAFE CDF. The difference between the two ranges from 0 - 0.2. For the test set, the predicted mean CDF initially slightly lies below the true CDF curve at lower values, becomes steeper and overestimates at mid-range, and alternates again at higher values. While these trends suggest some instability in the model's predictions for higher values, the absolute difference between the two CDFs remains within 0.1, indicating limited deviation. It is worth noting that the scatter plot in Fig. 6 shows the test mean NPP predictions distributed more evenly around the 1:1 line. This apparent discrepancy arises from the differing perspectives of the two plots: the CDF curve highlights cumulative differences across the distribution, whereas the scatter plot reflects point-wise deviations. Together, these visualizations suggest that while the model captures the overall distribution trends well, some localized errors in predicting mid-range and higher values may contribute to these patterns.

For the Bayesian probabilistic prediction model, the predicted mean CDF curve is above the true value in the training set. When the CAFE increases to a certain extent, the two curves alternate, and the absolute value of the difference between the CDF does not exceed 0.2. In the test set, the two CDF curves overlap first and then separate. The predicted mean CDF rises more quickly, and is on top of the true value CDF curve, with the difference between the two curves not exceeding 0.1 when the CAFE increases to a

certain extent. When the NPP increases to a certain degree, the two curves overlap again, and the absolute value of the difference between the CDF does not exceed 0.3. Overall, the difference between that of the predicted mean values and the CDF of the true values obtained by the two models is small, which indicates that the overall deviation of the model predictions is not large, and both models show good prediction performance and can capture the statistical characteristics of the data well. However, the CDF curves of the neural network probabilistic prediction model are closer to the true values on both the training and test sets, possibly implying that the neural network model is more effective in dealing with complex data and capturing nonlinear relationships. The flexibility of neural networks allows them to adapt to different data distributions and patterns.

Table 5 presents RMSD, MAPD, and CRPS for both models using CAFE as prediction target. Additionally, we analyzed the proportion of raw input data encompassed within the 95% confidence interval, thereby providing a more nuanced evaluation of the model's proficiency in capturing CAFE uncertainty. According to Table 5, the neural network-based probabilistic prediction model exhibits superior performance in terms of CRPS, RMSD, and MAPD. This denotes a higher level of accuracy and reliability for the neural network model in probabilistic predictions of CAFE, especially when considering uncertainty. Conversely, the Bayesian probabilistic prediction model demonstrates a stronger ability to encompass a greater proportion of the raw input data within the 95% confidence interval. This suggests that while it may exhibit higher overall uncertainty, it has a more pronounced capability to capture the nuances of uncertainty.

This comparative analysis elucidates that both the neural network-based probabilistic prediction model and the Bayesian probabilistic prediction model, grounded in empirical distributions, are adept at capturing and quantifying the uncertainty of CAFE. While the Bayesian model demonstrates a heightened capability in encompassing a broader scope of uncertainty, the neural network model distinguishes

itself by its superior accuracy and reliability, particularly in precisely predicting the uncertainty of CAFE. A notable observation is that when CAFE values exceed 350 mg $C m^{-2} d^{-1}$, the predictive performance of both models deteriorates. This manifests as an underestimation of mean predictions, indicating an inability to fully and accurately predict NPP across the entire range of size classes. The underlying reason for this may stem from the considerable variation in the input data and its skewed sample distribution. Most notably, a significant proportion of the samples were primarily concentrated within the 200-350 mg $C m^{-2} d^{-1}$ range. In contrast, CAFE values exceeding 350 mg $C m^{-2} d^{-1}$ constitute only 28% of the input dataset. Consequently, the models exhibit insufficient learning of higher value ranges during the training phase, resulting in a notable prediction bias for larger CAFE values.

## 3.3. Probabilistic Prediction of NPP in Weizhou Island (2007–2018)

Given the 8-day temporal resolution of data acquired by remote sensing satellites and the consequent data incompleteness, this study employed the previously trained neural network and the Bayesian probabilistic prediction models using CAFE as training target to forecast the daily NPP in the Weizhou Island sea area from 2007 to March 2018, thereby supplementing the NPP dataset. This approach aligns with the focus established in Section 3.1, which emphasizes the efficacy of probabilistic prediction models when CAFE is used as the prediction target. The selection of CAFE outputs reflects the model's relative strengths in representing phytoplankton-based NPP dynamics in the study area, as well as the high accuracy achieved by the NN and Bayesian models in emulating its output. The results are illustrated in Fig. 10, where the predicted mean values and 95% confidence intervals for both models are displayed. Fig. 10(c) reveals that the Bayesian model's confidence interval is broader, primarily due to its lower limit, yet no substantial difference is noted between the predicted mean values of the two models. Both models effectively mirror the trend of NPP. The analysis of the annual change of NPP shows a clear periodicity, which means that the change of NPP is not random, but follows certain laws and patterns. Combined with Fig. 11, the

seasonal variation of NPP throughout the year emerges. Specifically, NPP shows a
decreasing trend from January to July each year, with July generally being the lowest
level of the whole year. Then it increases from July to November and slightly decreases
from November to December. Overall, NPP has larger values in winter and spring.
These results provide important insights into seasonal variations and interannual trends
of NPP in the Weizhou Island waters and provide valuable data to support the study of
the marine ecosystem dynamics.
However, the significance of our work extends far beyond mere data replication.
The primary aim of our study is to enhance the reliability of marine NPP estimates by
using advanced probabilistic models. Our objective extends beyond merely reproducing
satellite NPP products. We aim to improve the overall accuracy and uncertainty
quantification of NPP estimates by incorporating a robust probabilistic framework. This
framework helps to better understand and quantify the uncertainties inherent in marine
NPP, whether they originate from satellite data or environmental factors. By using
Bayesian models and probabilistic neural networks, we not only replicate satellite NPP
estimates but also capture and quantify uncertainties at multiple levels. These models
account for uncertainties in the satellite products, input data variability, and the
predictive model itself, thus providing a more comprehensive uncertainty quantification
relevant to marine NPP.

## 4. Conclusion


This study primarily addresses the challenge of uncertainty in satellite ocean color
data estimates of ocean NPP. Departing from traditional point estimation regression
models, we embraced a probabilistic prediction approach where the output is a
probability distribution. The models utilized in this study include a Bayesian
probabilistic prediction model based on empirical distributions and a deep learning-
based probabilistic prediction model under the TFP framework. Focusing on the NPP
uncertainty analysis in the Weizhou Island sea area, we explored the effect of the
probabilistic prediction model when the NPPs obtained by the VGPM, CbPM, and

CAFE methods, respectively, are used as the prediction targets. Unlike traditional models such as VGPM, CbPM, and CAFE, the NN and Bayesian probabilistic models are designed to capture complex nonlinear interactions between environmental variables and NPP while providing robust uncertainty quantification. These probabilistic models do not require additional input variables beyond those used by VGPM, CbPM, and CAFE. Moreover, Pearson correlation analysis allows for the identification of the most critical inputs for prediction. By prioritizing variables such as SST and AP, the models can be optimized to reduce reliance on less influential inputs, improving efficiency without compromising accuracy. Furthermore, this study compares and analyzes the capabilities of Bayesian and neural network probabilistic models in predicting the CAFE uncertainty. The results reveal that both models are competent in quantifying CAFE uncertainty.

When exploring the uncertainty of the NPP using the Bayesian probabilistic prediction model and the neural network probabilistic prediction model, the results show that the two probabilistic prediction models are the most effective when the prediction target is the CAFE. The probability distributions obtained by the two probabilistic prediction models are similar to those of CAFE, with the difference in CDF between the predicted mean and true values at each data point not exceeding 0.2 for the neural network probabilistic prediction model and 0.3 for the Bayesian probabilistic prediction model. In contrast, the confidence intervals for the outputs of the Bayesian probabilistic prediction model are wider, and the proportion of the CAFE that falls in the confidence intervals is higher, which shows that Bayes is more capable of capturing uncertainty, but its accuracy is not high. However, the neural network probabilistic prediction model is more accurate and reliable. Its performance is better in many assessment indicators, but not all CAFE values in the size range can be predicted accurately by the model. When the CAFE is less than 450 mg C m$^{-2}$ d$^{-1}$, the model tends to overestimate the actual NPP value. When CAFE is larger than 600 mg C m$^{-2}$ d$^{-1}$, it tends to underestimate the actual NPP value. When the two probabilistic prediction models are applied to the prediction of CAFE in the Weizhou Island waters

between January 2007 and February 2018, the prediction results illustrate the interannual trend of CAFE, and the magnitude of NPP is found to show obvious cyclic changes. Our study demonstrates the novel application of advanced probabilistic models to the remote sensing of marine NPP. By addressing the uncertainties in satellite-derived estimates and improving the reliability of NPP predictions, our work contributes to advancing the field of marine remote sensing and provides a foundation for future research.

An important limitation of this study is that the probabilistic prediction models were trained on outputs from existing NPP models rather than directly on observational data. This introduces the potential for inherited biases and errors from the base models, limiting the generalizability of our uncertainty estimates to true NPP values. Future research should prioritize incorporating in situ NPP measurements to refine model training and validation, enabling more accurate and reliable uncertainty quantification. The differences between VGPM, CbPM, and CAFE outputs underscore the challenges in determining the most reliable NPP training data. While CAFE was chosen as the primary prediction target, this choice was informed by prior studies highlighting its strengths in parameterizing key oceanic processes and by the strong predictive performance of the NN and Bayesian models when using CAFE outputs. We acknowledge that this approach inherits the limitations of the base models and that further validation with in situ measurements is necessary to ensure that CAFE outputs align closely with true NPP values. While our approach demonstrates strong potential for accurately quantifying NPP uncertainty in this specific marine area, its application to larger regions may encounter scalability challenges. This limitation arises due to the large number of input variables required for the neural network and Bayesian probabilistic models, which necessitate significant computational resources and extensive observational data coverage.

In the context of ongoing climate change, accurately capturing and reducing the uncertainty of marine NPP emerges as a key research focus in marine ecology. This

endeavor is crucial for a deeper understanding of energy and matter flow in marine ecosystems, providing a solid scientific foundation for the judicious management of the conservation of natural resources. While our study has advanced the field by demonstrating the feasibility of probabilistic prediction in quantifying NPP uncertainty, we acknowledge the potential for further enhancements and expansions. Looking ahead, future research could embark on the following paths to augment our work: (1) Expanding the research scope: The current study has concentrated primarily on specific marine areas. Future initiatives could broaden this focus to encompass diverse geographic regions and types of marine ecosystems. However, such an expansion would require addressing the scalability limitations inherent to the current models, such as their reliance on a high volume of input variables and computational resources. Investigating strategies to simplify model inputs or develop hierarchical approaches that adapt to varying data availability and resolution across broader regions would be critical for enhancing scalability. This expansion is vital to gain a more comprehensive understanding of probabilistic prediction's applicability and effectiveness across varying environmental conditions; (2) Enhancing data collection: The acquisition of more extensive and comprehensive observational data is instrumental in refining model training and prediction accuracy. Future endeavors should aim to amass a richer array of observational data, emphasizing the need for long-term time series and high-resolution remote sensing data. These efforts will significantly bolster the development and validation of robust probabilistic prediction models; (3) Refining model structure: Our study utilized Bayesian probabilistic regression and deep learning-based probabilistic prediction models. Future studies could explore the integration of other advanced model structures or the optimization of the existing ones, aiming to elevate the model's performance and robustness. Through these concerted efforts, we aspire to continually refine the methodologies of probabilistic prediction in quantifying marine NPP uncertainty, thereby laying the groundwork for more precise ecosystem management and environmental protection strategies.

**Author contribution Statement**

Jie Niu: Conceptualization, Methodology, Data Curation, Writing - Review & Editing,
Supervision, Funding acquisition.
Mengyu Xie: Conceptualization, Methodology, Data Curation, Writing - Original Draft,
Visualization.
Yanqun Lu: Conceptualization, Methodology, Data Curation, Writing - Original Draft,
Visualization.
Liwei Sun:Data Curation, Supervision,Funding acquisition.
Na Liu: Writing - Review & Editing, Supervision.
Han Qiu: Writing - Review & Editing, Supervision.
Dongdong Liu: Writing - Review & Editing, Supervision.
Chuanhao Wu: Writing - Review & Editing, Supervision.
Pan Wu: Writing - Review & Editing, Supervision.

## Declaration of interests

The authors declare that they have no known competing financial interests or personal
relationships that could have appeared to influence the work reported in this paper.

## Acknowledgements

This research was funded by the National Natural Science Foundation of China
[41972244], the Project of Science and Technology Department of Guizhou Province
(the technical system of prevention and control to mine groundwater pollution in karst
areas, 2022), and the High-Level Talent Training Program in Guizhou Province
(GCC[2023]045). Co-author Liwei Sun was partially supported by GuangDong Basic
and Applied Basic Research Foundation (2022A1515010590).

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

Bayesian model averaging. Energy, 308, 132884.

## Tables

**Table 1.** Summary of Variables and Data Sources.

| Variable name | Variable description | Data source |
|---|---|---|
| SST | Sea surface temperature (℃) | |
| Sal | Salinity (‰) | |
| TH | Height of tide(m) | |
| AP | Air pressure (hPa) | Weizhou Marine environment |
| RH | Relative humidity (%) | monitoring station |
| SV | Sea visibility (km) | |
| WS | Wind speed (m·s$^{-1}$) | |
| H/10 | 1/10th significant wave height (m) | |
| PAR | Photosynthetically active radiation (W·m$^{-2}$) | Oceancolor |
| SSP | Sea surface precipitation (mm) | Earthdata |
| SH | Sunshine hours (h·d$^{-1}$) | China Meteorological Administration |
| VGPM | NPP from the VGPM model (mgC m$^{-2}$·d$^{-1}$) | |
| CbPM | NPP from the CbPM model (mgC m$^{-2}$·d$^{-1}$) | Ocean Productivity |
| CAFE | NPP from the CAFE model (mgC m$^{-2}$·d$^{-1}$) | |

**Table 2.** Summary of Missing Variables.

| Variable | SV (km) | H/10 (m) | PAR (W·m$^{-2}$) | SSP (mm) | SH (h·d$^{-1}$) |
|---|---|---|---|---|---|
| Missing quantity (days) | 31 | 51 | 828 | 378 | 18 |

**Table 3.** Statistics of data pre- and post-interpolation.

| | SV (km) | | H/10 (m) | | PAR (W·m$^{-2}$) | | SSP (mm) | | SH (h·d$^{-1}$) | |
|---|---|---|---|---|---|---|---|---|---|---|
| | pre- | post- | pre- | post- | pre- | post- | pre- | post- | pre- | post- |
| count | 4046 | 4077 | 4026 | 4077 | 3249 | 4077 | 3699 | 4077 | 4059 | 4077 |
| mean | 15.22 | 15.23 | 0.57 | 0.57 | 34.92 | 35.97 | 4.94 | 4.85 | 5.19 | 5.18 |
| std | 10.33 | 10.30 | 0.41 | 0.41 | 15.64 | 15.20 | 16.13 | 15.61 | 3.93 | 3.93 |

| | | | | | | | | | | |
|---|---|---|---|---|---|---|---|---|---|---|
| min | 0.00 | 0.00 | 0.00 | 0.00 | 1.20 | 1.20 | 0.00 | 0.00 | 0.00 | 0.00 |
| 25% | 7.00 | 7.00 | 0.30 | 0.30 | 22.19 | 24.14 | 0.00 | 0.00 | 0.80 | 0.80 |
| 50% | 12.00 | 12.00 | 0.50 | 0.50 | 36.03 | 36.87 | 0.00 | 0.00 | 5.60 | 5.60 |
| 75% | 25.00 | 25.00 | 0.70 | 0.70 | 47.58 | 48.49 | 1.30 | 1.50 | 8.90 | 8.80 |
| max | 50.00 | 50.00 | 4.00 | 4.00 | 61.13 | 61.13 | 280.40 | 280.40 | 12.6 | 12.6 |

**Table 4.** Parameters of the Neural Network Model

| | Hyper-parameters | |
|---|---|---|
| | Layer 1 | 64 |
| | Layer 2 | 32 |
| Layer Sizes | Layer 3 | 16 |
| | Layer4 | 16 |
| | Layer 5 | 2 |
| | Distribution Layer | Gaussian distribution |
| Epochs | 800 | |
| Learning Rate | 0.0001 | |
| Batch Size | 16 | |
| optimizer | Adam | |
| loss | Negative log likelihood | |

**Table 5.** CRPS, RMSD, MAPD, and proportion of input data within 95% confidence interval.

| | CRPS | | RMSD | | MAPD | | Proportion | |
|---|---|---|---|---|---|---|---|---|
| | Train | Test | Train | Test | Train | Test | Train | Test |
| NN | 0.096 | 0.133 | 0.149 | 0.198 | 11.828 | 13.237 | 0.971 | 0.932 |
| Bayes | 0.151 | 0.20 | 0.201 | 0.253 | 13.909 | 14.145 | 0.976 | 0.951 |

940

## Figures

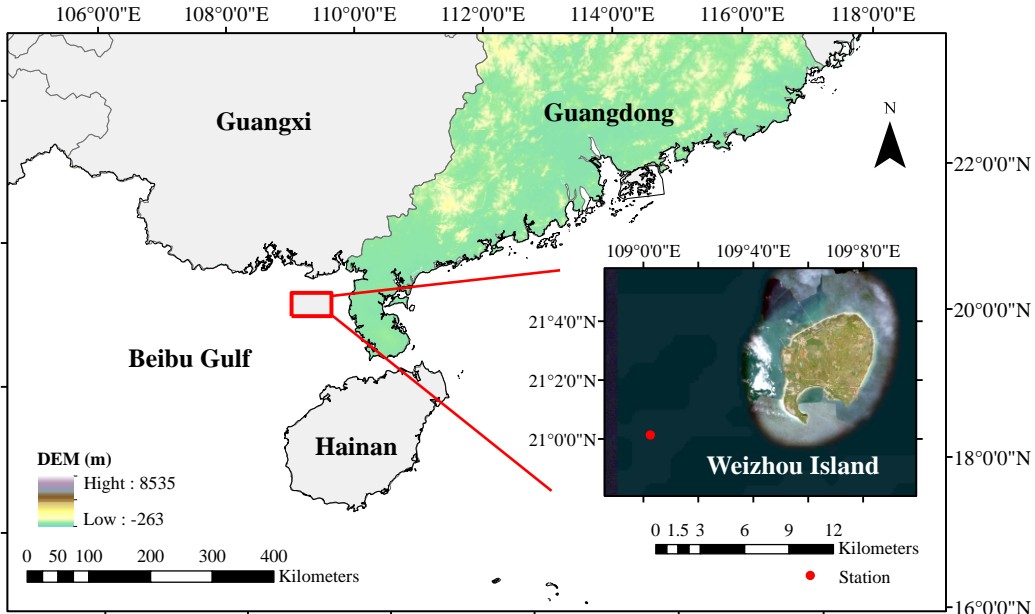

**Fig. 1.** The research area is located in the waters of Weizhou Island in Beibu Gulf, south China. The red dots in the figure indicate the location of Weizhou Marine Environmental Monitoring Station (21.0017°N, 109.0117°E). Eight distinct sets of monitoring data were collected from this monitoring station.

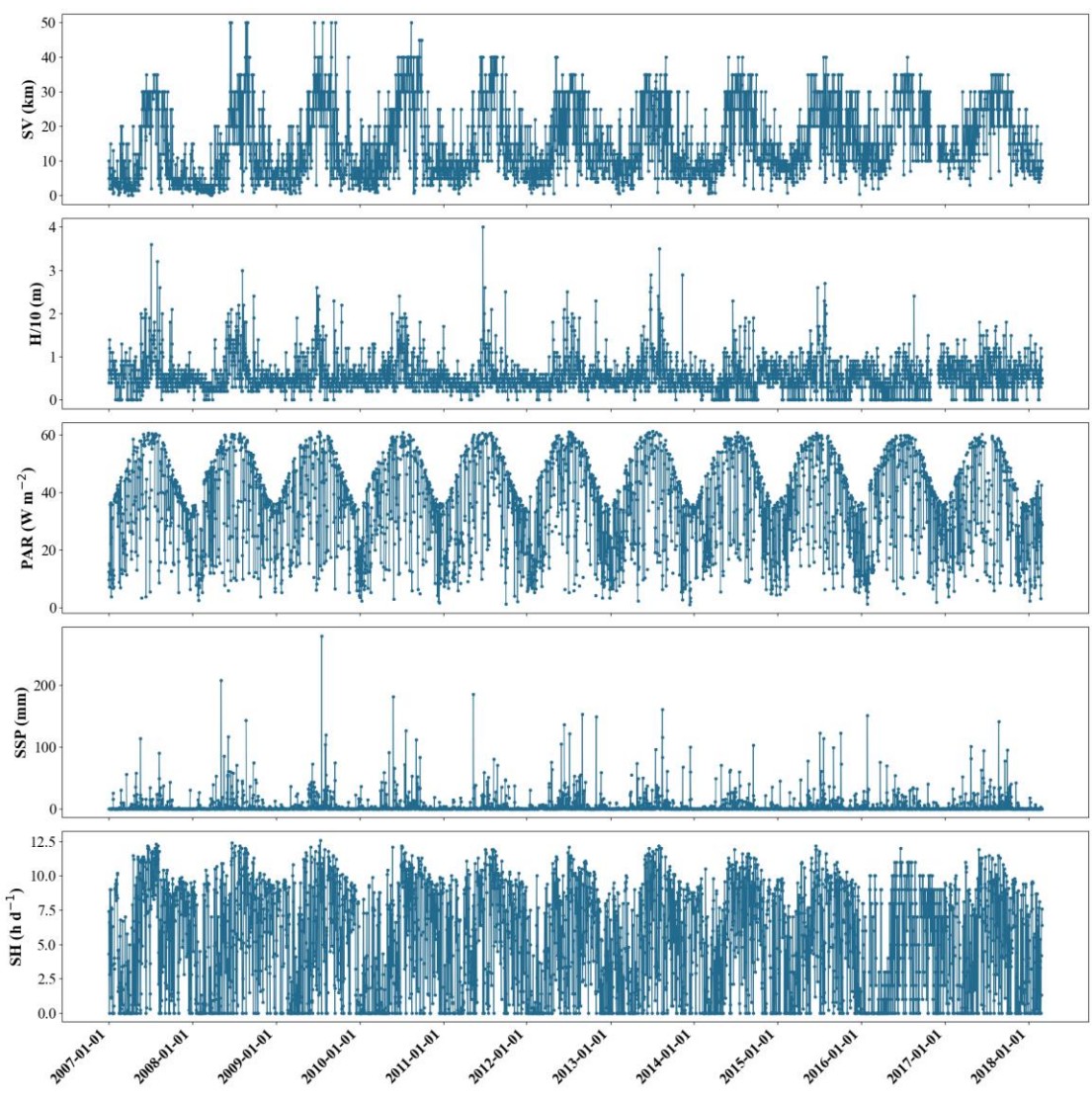


**Fig. 2.** Time series plots of SV, H/10, PAR, SSP, and SH with missing variables, showing the cyclical variation of these five variables.



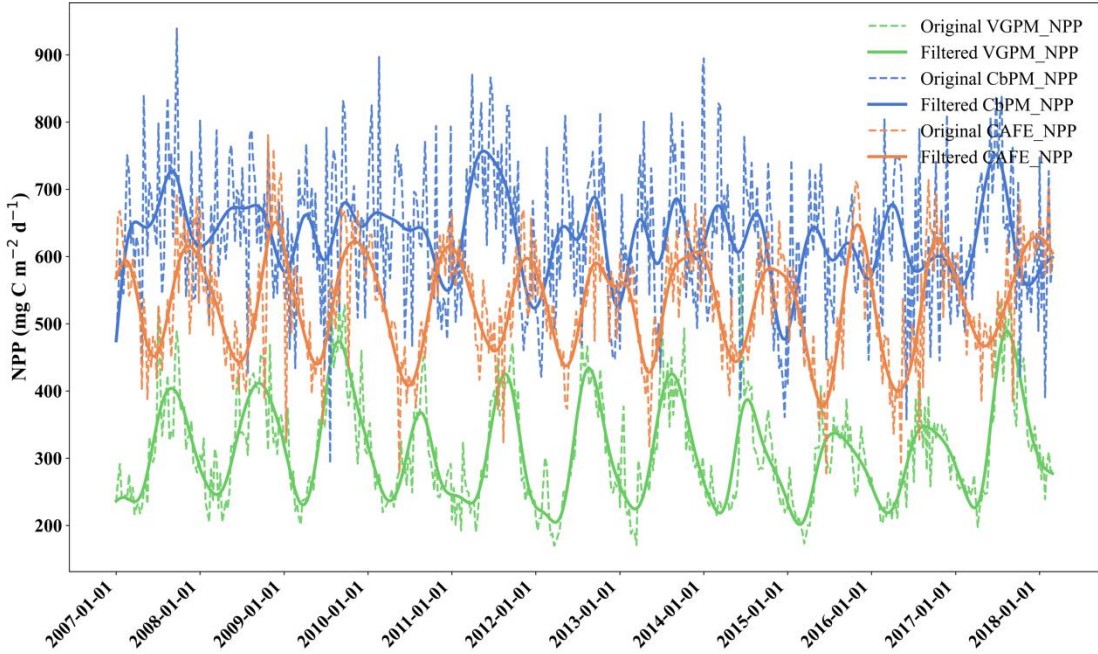


**Fig. 3.** Time series of VGPM, CbPM, and CAFE from January 2007 to February 2018, where the green line represents VGPM, the blue line represents CbPM, and the orange line represents CAFE . The dashed lines are the original data and the solid ones are the low-pass filtered, which show the seasonal variations more clearly. Abbreviations and data sources can be referenced in Table 1.

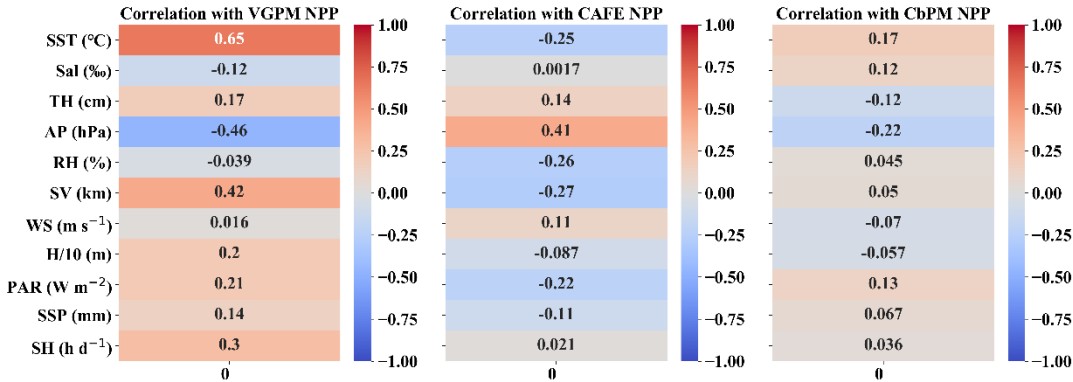

956

**Fig. 4.** Pearson correlation between the 11 input variables and the three NPPs (VGPM, CAFE, and CbPM). These input variables serve as inputs to the probabilistic models, while VGPM, CAFE, and CbPM are used as model outputs. The deeper the shade of red indicates a stronger positive correlation, whereas the deeper shade of blue indicates a stronger negative correlation.

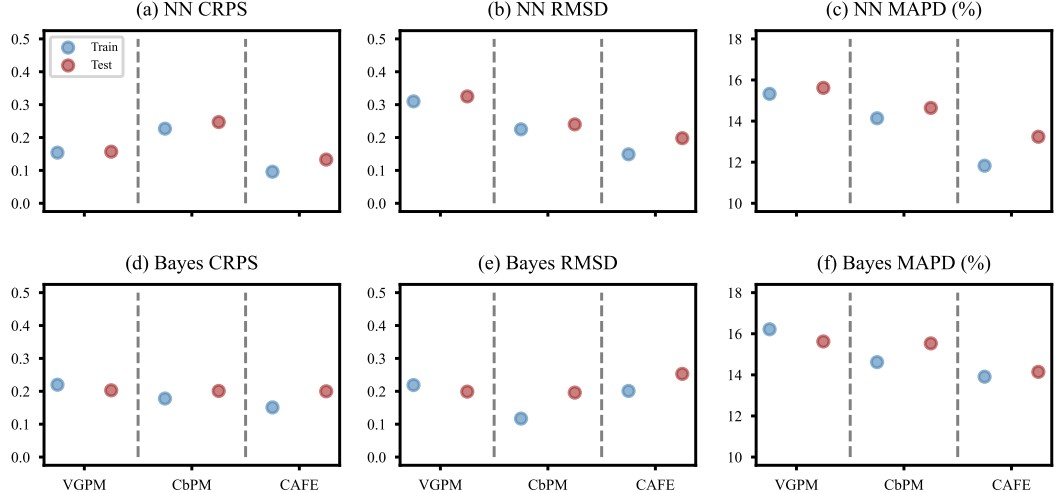

961

**Fig. 5.** Comparison of NPP predictive effects from VGPM, CbPM, and CAFE. Panels (a)–(c)
present the results from the neural network-based probabilistic prediction models; panels (d)–(f) the
results from Bayesian probabilistic prediction models based on empirical distributions. The
horizontal coordinates represent the VGPM, CbPM, CAFE as inputs in sequence, separated by gray
dashed lines, where blue dots represent data from the training set, and red dots denote data from the
test set, and the vertical coordinates are the values of the three metrics, CRPS, RMSD, MAPD. Since
NPP values were normalized to the range of $0 - 1$, the y axes of subplots (a), (b), (d), and (e) are
dimensionless. The units for MAPD are percentile.

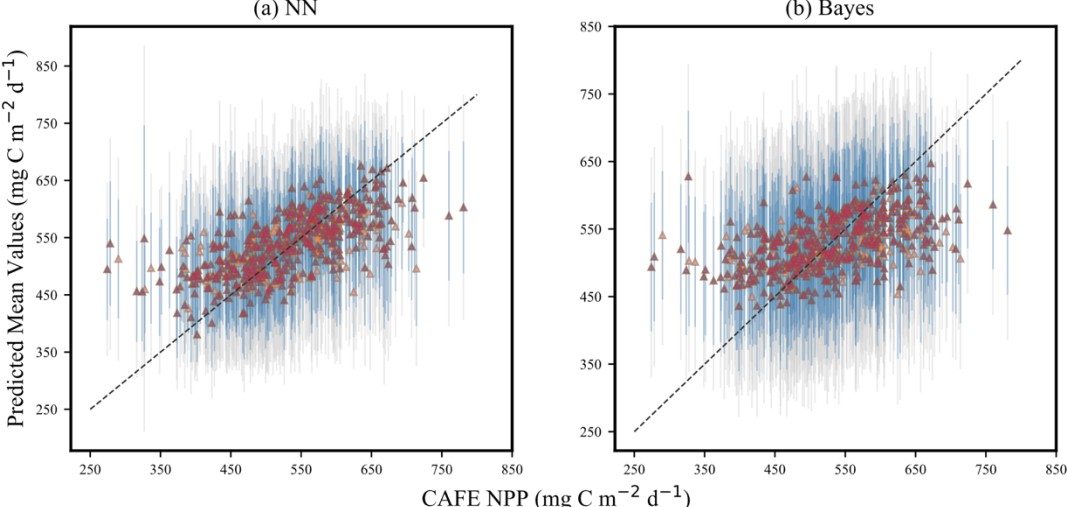

970

**Fig. 6.** Uncertainty quantification of (a) neural network-based probabilistic prediction model and (b)
empirical distribution-based Bayesian probabilistic prediction model. The horizontal axes represent
the input CAFE value, while the vertical axes show the mean predicted by the model. The triangular
icons in the figure represent 514 sets of the forecast average, the gray vertical lines represent the
95% confidence intervals for the predictions, and the blue vertical lines represent the 75%
confidence intervals.

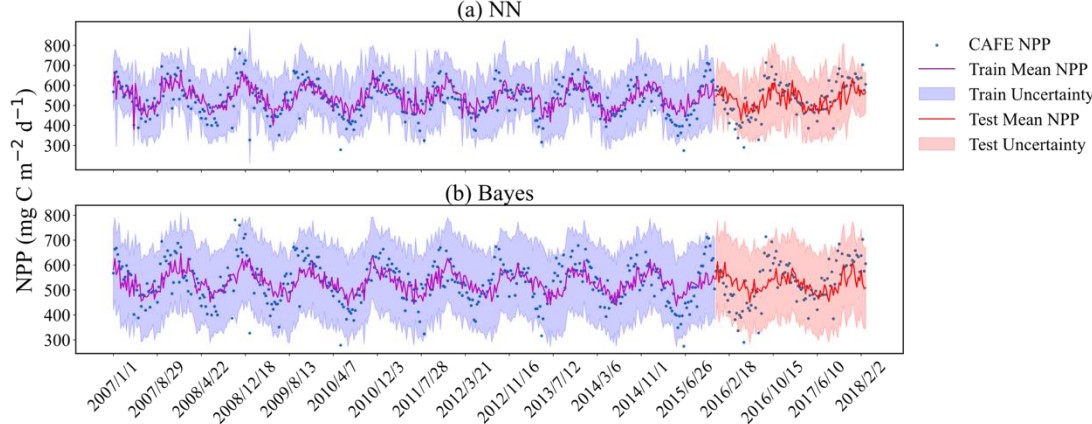

977

**Fig. 7.** Comparison of original and predicted mean values shown at an 8-day temporal resolution
within a 95% confidence interval. (a) Probabilistic prediction results based on neural networks; (b)
Bayesian probabilistic prediction results based on empirical distributions. The dashed lines
represent the mean values of the probabilistic predictions. The purple and red shaded areas illustrate
the uncertainty ranges for the training and the test sets, respectively. Blue dots signify observed data
points. All predictions and observations are presented in chronological sequence.

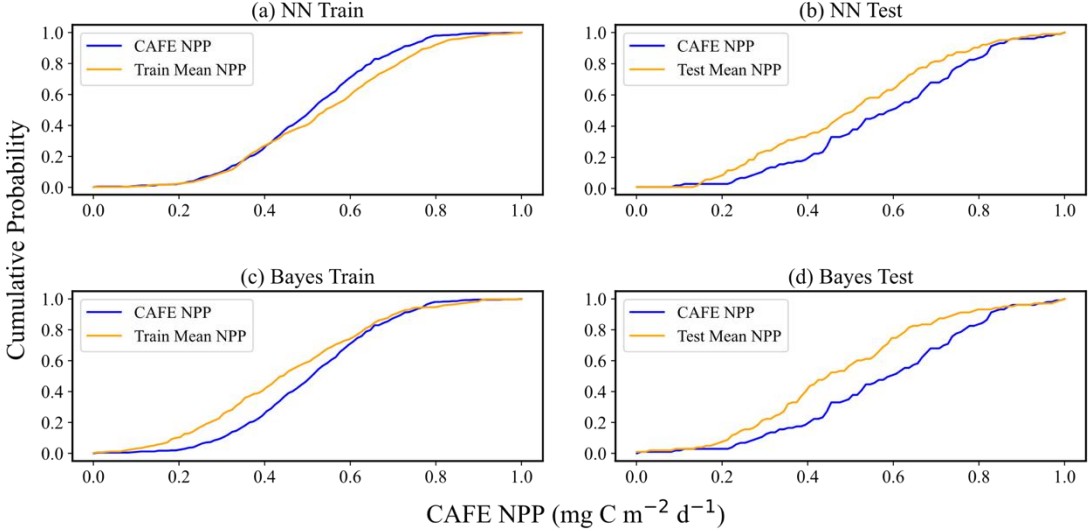

984

**Fig. 8.** Comparison of CAFE and predicted mean CDF. Panels (a) and (b) display the performance
of the training and test sets, respectively, in the neural network-based probabilistic prediction model.
Panels (c) and (d) illustrate the performance of the training and test sets, respectively, in the
empirical distribution-based Bayesian probabilistic prediction model. The data has been normalized
to a scale of 0–1 to ensure consistency across metrics and facilitate direct comparison between the
two models. In each panel, the blue curves represent the CDFs of the CAFE values, while the yellow
curves depict the CDFs of the model's predicted mean values.

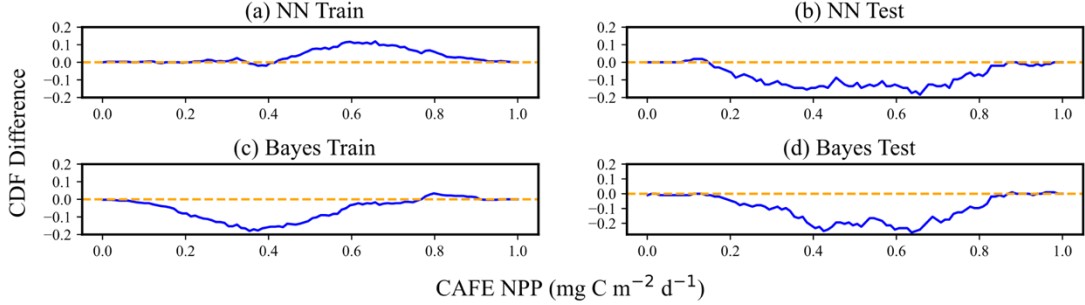

992

**Fig. 9.** Difference between the CAFE CDF and predicted mean CDF of model predictions. Panels (a) and (b) represent the performance of the training set and test sets, respectively, in the neural network-based probabilistic prediction model. Panels (c) and (d) showcase the performances of the training set and test sets, respectively, in the empirical distribution-based Bayesian probabilistic prediction model. The residuals are expressed in normalized units (0–1), enabling consistent assessment of model performance across different NPP ranges. The blue curves in each panel indicate the differential magnitude of the CDFs. Instances where the blue curves align with the yellow lines denote zero discrepancy between the input data CDF and the model's predicted mean CDF.

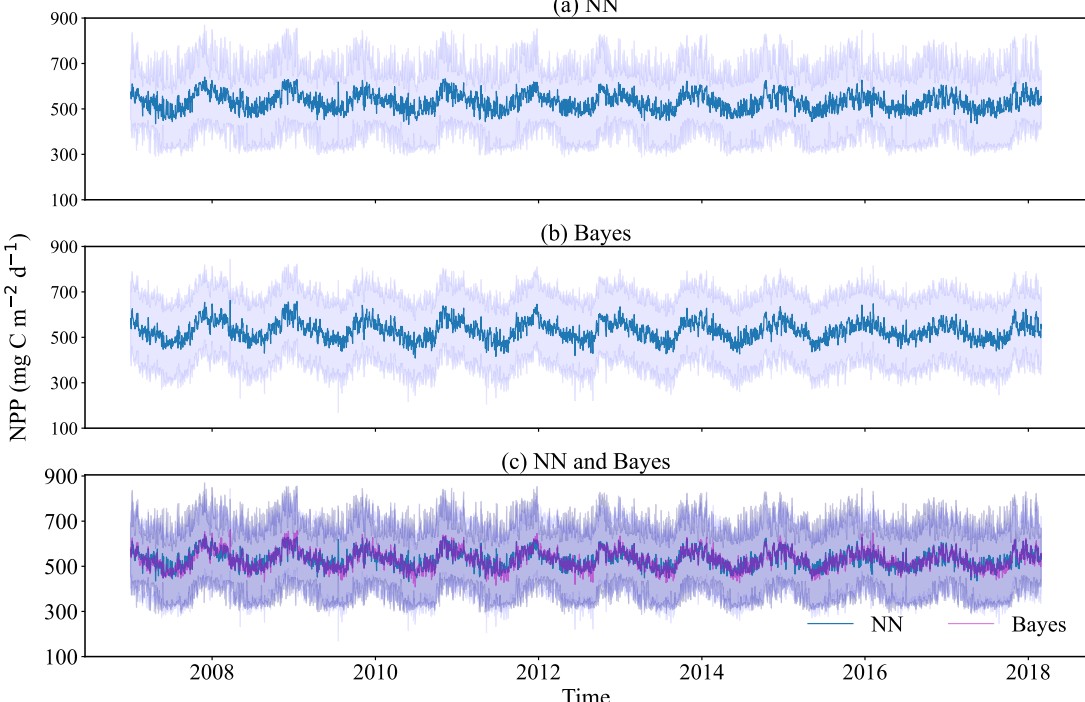

1002

**Fig. 10.** Time series plots of daily probabilistic NPP predictions in Weizhou Island (2007 – March 2018). (a) Probability prediction results of the neural network model; (b) Bayesian probability prediction results based on empirical distribution; (c) Comparison of the two models' predictions, with the green lines representing the mean predictions from the neural network model and the gray lines depicting the mean predictions from the Bayesian model.

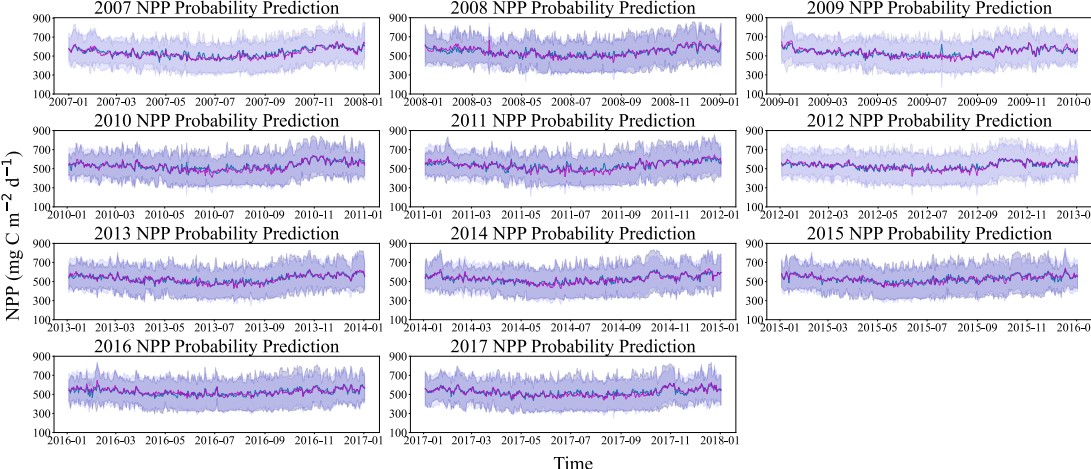

1008

**Fig. 11.** Time series plots of probabilistic NPP predictions in Weizhou Island (2007 – 2017). The light purple shading indicates the 95% confidence interval of the Bayesian model, while the dark purple shading represents the 95% confidence interval of the neural network model. The green lines show the mean prediction values from the neural network model; and the gray lines depict the mean prediction values from the Bayesian model.