# Peer review of "Refining Marine Net Primary Production Estimates: Advanced"

_EGUsphere, 2024_

## Referee Comment (RC1)

**Detailed Review for 'Refining Marine Net Primary Production Estimates: Advanced Uncertainty Quantification through Probability Prediction Models', Jie Niu et al.**

1. Line 26: In the abstract, the source of the NPP estimate (i.e., model output or observation) used in the paper should be mentioned.

2. Line 30: The author should explain the nature and the sources of uncertainty in NPP estimates. And why it is important.

3. Line 61-61: It is important to mention the recent study in Satyendranath et al. 2020 (Reconciling models of primary production and photoacclimation, Applied Optics)

4. Line 122: Again, it's important to mention why estimating uncertainty is important?

5. Line 137-138: Authors should rephrase "discloses the results" to "discusses the results".

6. Line 167-167: Why are these variables (input features) important in terms of estimating NPP?

7. Line 164: For PAR, SSP, SH and NPP data, authors should mention direct links for the data they used for experiments.

8. Line 783: Table-1: No need to mention the links here, acronyms are sufficient.

9. Line 785: Table-2: Authors should be more clear about the "missing quantity" units i.e., days.

10. Line 187: What specific algorithm was applied to make the time series interpolation?

11. Line 198-216: Authors can drop using "NPP" repeatedly, just the algorithm name is sufficient.

12. Line 208-211: It is not clear why CbPM is negatively correlated with AP. Authors should give an explanation.

13. Line 223: Typo in equation number.

14. Line 282: It is not clear whether the author had normalised the input features since they are in different scales.

15. Line 378: Do the authors have any explanation behind finding the lowest CPRS value than the other models?

16. Line 466-467: Applying a low pass filter on the time series is recommended before reaching this conclusion about long-term trend.

17. Line 478-481: Any previous studies (reference papers) that can support the statement about Bayesian model performing better in estimating uncertainty?

18. Line 483: What formula did the authors use to estimate the CDFs?

19. Line 486-487: As mentioned in the previous comment, the estimation of Train mean NPP and CAFE NPP curves are not clearly mentioned.

20. Line 505 "Small" should be replaced by "lower values" for more clarity.
21. Line 509-515: Test mean NPP lying below at lower values and the alteration at higher values is not appearing very significantly. Also, test mean NPP seems to over-estimate at mid-range but this is not the same as seen in the scatter plot (Fig. 6) where it is almost evenly distributed across either side of the 1:1 line.
22. Fig 10: The curves are difficult to distinguish. Different choice of colours recommended.
23. Fig 10: Capturing the seasonal cycle is fairly easy as most of the input features contain the same signal. To have a better understanding about how good the models are in reproducing the extreme values, authors should plot the anomaly time series by removing seasonal signals overlayed with observation treated in the same way.

---

## Author Response (AR1)

Dear Editor and anonymous Reviewers,

We express our sincere gratitude for the insightful comments and constructive criticisms on our manuscript titled "*Refining marine net primary production estimates: Advanced uncertainty quantification through probability prediction models*" (MS No.: egusphere-2024-3221). In response to your valuable feedback, we have meticulously revised our manuscript to enhance its clarity, coherence, and overall scientific contribution. Specific modifications have been made to address each point raised by the reviewers, and these are detailed in the subsequent pages, where we provide a point-by-point response to your comments. Reviews' comments are in normal text, whereas our responses are in blue.

This revision process has been a collaborative effort among all co-authors, and we believe that the adjustments made significantly improve the manuscript. We are confident that these changes have addressed your concerns and enriched the manuscript.

With kind regards,

Mengyu Xie (on behalf of all co-authors)

**Reviewer #1:**
Detailed Review for '*Refining Marine Net Primary Production Estimates: Advanced Uncertainty Quantification through Probability Prediction Models*', Jie Niu et al.

1. Line 26: In the abstract, the source of the NPP estimate (i.e., model output or observation) used in the paper should be mentioned.

We have revised the abstract to explicitly mention the source of the NPP estimate and have provided details about the research location and data used in the study (Lines 28-34).

"This study focuses on the aquatic environs of Weizhou Island, located off the coast of Guangxi, China, and introduces an advanced probability prediction model aimed at improving NPP estimation accuracy while addressing its associated uncertainties. The dataset comprises eleven distinct sets of monitoring data and satellite data spanning from January 2007 to February 2018. NPP values were derived using three widely

recognized estimation methods — VGPM, CAFE, and CbPM — serving as model outputs for further analysis."

2. Line 30: The author should explain the nature and the sources of uncertainty in NPP estimates. And why it is important.

We have revised the abstract to clarify the sources and nature of uncertainty in NPP estimates and to emphasize their significance. Specifically, we have included information about the challenges arising from measurement difficulties, errors in satellite-based inversion, and the need for reliable uncertainty quantification to improve ecosystem management and global carbon cycle modeling (Lines 23 - 28).

"In marine ecosystems, Net Primary Production (NPP) is important, not merely as a critical indicator of ecosystem health, but also as an essential component in the global carbon cycling process. Despite its significance, the accurate estimation of NPP is plagued by uncertainty stemming from multiple sources, including measurement challenges in the field, errors in satellite-based inversion methods, and inherent variability in ecosystem dynamics."

3. Line 61-61: It is important to mention the recent study in Satyendranath et al. 2020 (Reconciling models of primary production and photoacclimation, Applied Optics)

We thank the reviewer for highlighting this relevant study. In response, we have incorporated a reference to Satyendranath et al. (2020) into the revised manuscript. Specifically, we have added a sentence to emphasize their contribution to improving primary production models by addressing parameter assignment and its impact on reducing uncertainties (Lines 77 - 80).

"Satyendranath et al. (2020) emphasize the critical role of accurately assigning parameters in primary production models as a key strategy for reducing model uncertainties and enhancing the reliability of satellite-based primary production estimates, particularly in the context of climate research."

4. Line 122: Again, it's important to mention why estimating uncertainty is important?

In the revised manuscript (Lines 129 - 135), we have included sentences to emphasize the significance of uncertainty estimation.

"The estimated values of NPP derived from the above three classical models exhibit significant discrepancies, reflecting substantial uncertainties in these methods. These inaccuracies can impede a comprehensive understanding of the role of oceans in the global climate system, particularly in their capacity to act as carbon sinks and regulators of atmospheric CO2 levels. Consequently, quantifying and addressing these uncertainties is primary to improving the reliability of NPP estimates and ensuring their applicability in climate research and marine ecosystem management."

5. Line 137-138: Authors should rephrase "discloses the results" to "discusses the results".

Corrected (Line 153).

" Section 4 discusses the results; and Section 5 presents the conclusions."

6. Line 167-167: Why are these variables (input features) important in terms of estimating NPP?

We have added detailed explanations in the revised manuscript to clarify the relevance and importance of the input variables for estimating NPP (Lines 182 - 191).

"These data were aggregated to constitute a comprehensive dataset encompassing eleven variables, serving as the input features for the models. Phytoplankton, the primary source of NPP, is directly influenced by variables such as SST, Par, and SH, which are critical to its photosynthetic processes. Additionally, other variables have significant indirect effects on phytoplankton growth. Sal, for example, influences the community structure of phytoplankton (Braarud et al., 1951). Variables such as TH,

H/10, and WS indirectly affect phytoplankton dynamics by modulating water column mixing and the vertical distribution of nutrients. AP, RH and SV also indirectly impacts phytoplankton photosynthetic activity by altering environmental conditions."

7. Line 164: For PAR, SSP, SH and NPP data, authors should mention direct links for the data they used for experiments.

We have provided direct links to the datasets used in this study (Lines 179-182, Lines 195 - 197).

"For the analysis of three NPP algorithms—namely, VGPM, CbPM, and CAFE—we utilized their output datasets, which were obtained at an eight-day temporal resolution from the Ocean Productivity website (http://orca.science.oregonstate.edu/1080.by.2160.monthly.hdf.vgpm.m.chl.m.sst.php, http://orca.science.oregonstate.edu/1080.by.2160.monthly.hdf.cbpm2.m.php, http://orca.science.oregonstate.edu/1080.by.2160.monthly.hdf.cafe.m.php)."

8. Line 783: Table-1: No need to mention the links here, acronyms are sufficient.

We have corrected Table 1 by removing the dataset links and retaining only the acronyms, as suggested.

**Table 1.** Summary of Variables and Data Sources.

| Variable name | Variable description | Data source |
| --- | --- | --- |
| SST | Sea surface temperature (℃) | |
| Sal | Salinity (‰) | |
| TH | Height of tide(m) | |
| AP | Air pressure (hPa) | Weizhou Marine environment monitoring station |
| RH | Relative humidity (%) | |
| SV | Sea visibility (km) | |
| WS | Wind speed (m·s⁻¹) | |

| H/10 | 1/10th significant wave height (m) | |
| PAR | Photosynthetically active radiation (W·m⁻²) | Oceancolor |
| SSP | Sea surface precipitation (mm) | Earthdata |
| SH | Sunshine hours (h·d⁻¹) | China Meteorological Administration |
| VGPM | NPP from the VGPM model (mgC m⁻²·d⁻¹) | |
| CbPM | NPP from the CbPM model (mgC m⁻²·d⁻¹) | Ocean Productivity |
| CAFE | NPP from the CAFE model (mgC m⁻²·d⁻¹) | |

9. Line 785: Table-2: Authors should be more clear about the "missing quantity" units i.e., days.

Thank you for your reminder. We have updated Table 2 to include the unit "days" for the "missing quantity" column.

**Table 2.** Summary of Missing Variables.

| Variable | SV (km) | H/10 (m) | PAR (W·m⁻²) | SSP (mm) | SH (h·d⁻¹) |
|---|---|---|---|---|---|
| Missing quantity (days) | 31 | 51 | 828 | 378 | 18 |

10. Line 187: What specific algorithm was applied to make the time series interpolation.

In our research, we used the 'interpolate' function from the Python Pandas library, configured with the 'time' method, to perform the time series interpolation. This approach, while classified as linear interpolation, incorporates the time factor, ensuring that the intervals between timestamps are explicitly considered. This feature enhances its suitability for time series data, particularly datasets with periodic variations like those in our study, enabling more accurate estimation of missing values. Although it is computationally simpler than periodic interpolation methods (e.g.,

Fourier transform or time series models with seasonal decomposition), the 'time' method sufficiently captures the periodicity and variations inherent in our dataset, making it both efficient and effective for this application (Line 224-228).

" In this study, interpolation was used to address missing variables, and we ensure that the statistical properties of the original data were preserved to the greatest extent possible. This approach allows us to maintain the integrity of our analyses while recognizing the inherent limitations of using interpolated data."

11. Line 198-216: Authors can drop using "NPP" repeatedly, just the algorithm name is sufficient.

Corrected.

12. Line 208-211: It is not clear why CbPM is negatively correlated with AP. Authors should give an explanation.

Thank you for raising this insightful question. In response, we have elaborated on the relationship between AP and CbPM in the revised manuscript, providing an explanation for the observed negative correlation (Lines 260 - 264).

"Changes in AP affect atmospheric stability, cloudiness, and precipitation, indirectly altering light conditions in the ocean and subsequently affecting phytoplankton photosynthesis. Lower AP often corresponds to unstable atmospheric conditions and increased cloud cover, which may inhibit photosynthesis activity by reducing light penetration."

13. Line 223: Typo in equation number.

Corrected. (Line 275)

14. Line 282: It is not clear whether the author had normalised the input features since they are in different scales.

At the beginning of Section 2.3 of the article, it has been clarified that the input data of different scales have been normalized (Line 341).

"Prior to model evaluation, we normalized the NPP satellite data."

15. Line 378: Do the authors have any explanation behind finding the lowest CPRS value than the other models?

In the revised manuscript (Section 3.2.2, Lines 459 – 470, and 476 – 485), we have elaborated on potential factors contributing to the lower CRPS value for the CAFE model, in terms of both variance and cumulative distribution function. Also, Figs. S1 to S6 have been added in the SI to better explain the differences among the training and testing datasets of three NPPs.

[revised manuscript text omitted]

16. Line 466-467: Applying a low pass filter on the time series is recommended before reaching this conclusion about long-term trend.

We have applied a low-pass filter to the time series data for the three NPPs to isolate the long-term trends. The filtered results have been included in the revised Figure 3 to visually represent the smoothed trends, ensuring the analysis and conclusions are supported by appropriately processed data (Lines 237 – 243).

"To evaluate the long-term trends in Net Primary Production (NPP), we applied a low-pass filter to the three NPP products (VGPM, CbPM, and CAFE) (Fig. 3). This filtering process removes high-frequency variations, such as noise and short-term fluctuations, while retaining the underlying long-term patterns. It became evident that each exhibits a distinct seasonal periodicity, with the fluctuation ranges remaining stable over time yet the magnitude and timing of them varing significantly among the three NPPs."

[Figure]

**Fig. 3.** Time series of VGPM, CbPM, and CAFE from January 2007 to February 2018, where the green line represents VGPM, the blue line represents CbPM, and the orange line represents CAFE . The dashed lines are the original data and the solid ones are the low-pass filtered, which show the seasonal variations more clearly. Abbreviations and data sources can be referenced in Table 1.

17. Line 478-481: Any previous studies (reference papers) that can support the statement about Bayesian model performing better in estimating uncertainty?

Thank you for the comment. In the section introducing the Bayesian method (Lines 270 – 275), we have added citations to relevant literature to support the statement about the Bayesian model's superior performance in estimating uncertainty (lines 274– 275).

"Bayesian models can adeptly quantify the uncertainty in the distribution of predicted outcomes. The Bayesian approach is particularly advantageous in scenarios with limited training data or when potential invisibility in training data cannot be discounted in practical applications (Perfors et al, 2011; Kaplan D, 2021; Zou et al, 2024)."

18. Line 483: What formula did the authors use to estimate the CDFs?

In the revised manuscript, we have added a detailed explanation of the formula used to estimate CDFs (Lines 396 - 407).

"The Cumulative Distribution Function (CDF), also known as the distribution function, is the integral of probability density function (PDF). It provides a complete description of the probability distribution of a real-valued random variable $X$. The CDF is defined as the probability $P$ that a random variable $X$ is less than or equal to a given value $x$, expressed as:

$$F(x) = P(X \leq x)$$

To evaluate the predictive performance of the model, we computed the empirical CDF of the input data and compared it with the average predictive CDF generated by the model. This comparison provides a graphical representation of the model's predictive accuracy. A higher degree of overlap between the empirical and predictive CDF curves indicates a greater similarity between the two distributions, thereby reflecting superior model predictions."

19. Line 486-487: As mentioned in the previous comment, the estimation of Train mean NPP and CAFE NPP curves are not clearly mentioned.

Thank you for highlighting this point. In the revised manuscript, we have clarified that since our models generate probabilistic predictions, the curves presented in some figures represent the mean of these predictions. This clarification has been added in Section 2.3 to ensure transparency regarding the methodology and interpretation of the results (Lines 355 - 360).

"In this study, our models provide probabilistic predictions, generating a probability distribution for each time point rather than a single point estimate. To facilitate visualization and interpretation, the curves presented in some figures represent the mean values derived from these predictive distributions. These mean curves

summarize the central tendency of the model outputs while inherently accounting for the uncertainty associated with the predictions."

20. Line 505 "Small" should be replaced by "lower values" for more clarity.

Corrected. (Line 625)

21. Line 509-515: Test mean NPP lying below at lower values and the alteration at higher values is not appearing very significantly. Also, test mean NPP seems to over-estimate at mid-range but this is not the same as seen in the scatter plot (Fig. 6) where it is almost evenly distributed across either side of the 1:1 line.

We appreciate the reviewer's detailed observation. In response, we have revised the text to provide a clearer explanation of the observed patterns in the CDF curves and their relationship to the scatter plot (Fig. 6). Additionally, we have clarified the interpretation of the differences between the predicted and true value CDFs and provided insights into potential reasons for these discrepancies (Lines 623 - 635).

"As CAFE increases, the difference between the predicted and true CDF curves grows larger, with the predicted mean CDF on the training set generally lying below the CAFE CDF. The difference between the two ranges from 0 - 0.2. For the test set, the predicted mean CDF initially slightly lies below the true CDF curve at lower values, becomes steeper and overestimates at mid-range, and alternates again at higher values. While these trends suggest some instability in the model's predictions for higher values, the absolute difference between the two CDFs remains within 0.1, indicating limited deviation. It is worth noting that the scatter plot in Fig. 6 shows the test mean NPP predictions distributed more evenly around the 1:1 line. This apparent discrepancy arises from the differing perspectives of the two plots: the CDF curve highlights cumulative differences across the distribution, whereas the scatter plot reflects point-wise deviations. Together, these visualizations suggest that while the

model captures the overall distribution trends well, some localized errors in predicting mid-range and higher values may contribute to these patterns."

22. Fig 10: The curves are difficult to distinguish. Different choice of colours recommended.

We appreciate the reviewer's detailed observation. We have revised the colors in Fig 10 to better present the detailed information clearly. However, the contrast is not significant due to the fact that the predicted means of the two models are closer and the folds in the graph overlap more.

[Figure]

**Fig. 10.** Time series plots of daily probabilistic NPP predictions in Weizhou Island (2007 – March 2018). (a) Probability prediction results of the neural network model; (b) Bayesian probability prediction results based on empirical distribution; (c) Comparison of the two models' predictions, with the green lines representing the mean predictions from the neural network model and the gray lines depicting the mean predictions from the Bayesian model.

23. Fig 10: Capturing the seasonal cycle is fairly easy as most of the input features contain the same signal. To have a better understanding about how good the models are in reproducing the extreme values, authors should plot the anomaly time series by removing seasonal signals overlayed with observation treated in the same way.

Thank you for highlighting this point. We have drawn anomaly time series plot with seasonal signals removed (Figs. S7 and S8), and compared the ability of two probability prediction models to reproduce extreme values.

"To better understand the model's ability to reproduce extreme values, this article removed the seasonal signals from the original CAFE values and the predicted means of the two probabilistic prediction models and plotted the abnormal time series graphs (Figs. S7 and S8). From Fig. S7, it can be seen that the NN predicted mean values overlap more with the original values, better reflecting the fluctuation size of the original CAFE values, and is superior to Bayes in reproducing extreme values. Fig. S8 compares the prediction means of NN and Bayes when removing seasonal signals. As can be seen from the figure, when the models are applied to the NPP forecast from 2007 to March 2018, the average predictions of the two models are mostly close, but the NN output results fluctuate more significantly, better reflecting the complexity of the actual data."

[Figure]

Fig. S7. Comparison of CAFE and predicted mean values shown at an 8-day temporal resolution within a 95% confidence interval. In this case, the seasonal signals have been removed from the

original data and the predicted mean values to form an anomalous time series.

[Figure]

Fig. S8. Time series plots of daily probabilistic NPP predictions in Weizhou Island (2007 – March 2018). In this case, the seasonal signals have been removed from the predicted mean values to form an anomalous time series.

**Reviewer #2:**

The manuscript presents a comparative analysis of Bayesian and neural network-based probability prediction models for estimating Net Primary Production (NPP) at a location near Weizhou Island (though this spatial focus is not clearly stated in the abstract or introduction). While the study demonstrates interesting methodological approaches to uncertainty quantification, it requires major revisions and clarifications.

**general comments**

The spatial scope and context of the study need to be clearly defined in the abstract and introduction. The location or spatial extent of the study is not mentioned in the title, abstract or introduction, suggesting a global analysis of marine NPP, when in fact the study focuses on a specific (point) location near Weizhou Island off the Chinese coast. Given the large number of inputs required for the Neural Network (NN) and Bayesian technique used in the study, it would not be easy to scale the approach to a larger region.

Response: We thank the reviewer for this constructive comment. In response, we have revised the abstract and introduction to clearly define the spatial scope and context of the study. Specifically, we have included the geographic focus on the aquatic ecosystem near Weizhou Island, located off the Chinese coast, to ensure transparency and accuracy in presenting the study's scope (Lines 28 - 32). Additionally, we have addressed the scalability limitations of the proposed approach in the discussion section (Lines 786 – 799) to provide a balanced view of the study's applicability.

"This study focuses on the aquatic environs of Weizhou Island, located off the coast of Guangxi, China, and introduces an advanced probability prediction model aimed at improving NPP estimation accuracy while addressing its associated uncertainties. The dataset comprises eleven distinct sets of monitoring data and satellite data spanning from January 2007 to February 2018."

"While our study has advanced the field by demonstrating the feasibility of probabilistic prediction in quantifying NPP uncertainty, we acknowledge the potential for further enhancements and expansions. Looking ahead, future research could embark on the following paths to augment our work: (1) Expanding the research scope: The current study has concentrated primarily on specific marine areas. Future initiatives could broaden this focus to encompass diverse geographic regions and types of marine ecosystems. However, such an expansion would require addressing the scalability limitations inherent to the current models, such as their reliance on a high volume of input variables and computational resources. Investigating strategies to simplify model inputs or develop hierarchical approaches that adapt to varying data availability and resolution across broader regions would be critical for enhancing scalability. This expansion is vital to gain a more comprehensive understanding of probabilistic prediction's applicability and effectiveness across varying environmental conditions;"

A critical limitation of the study is the data used for training the NN and the Bayesian model. The models are trained on outputs from existing NPP models (VGPM, CbPM, and CAFE) rather than directly on NPP data. Effectively, the NN and Bayesian model serve as emulators of the NPP models, inheriting their underlying errors and biases. Thus, the uncertainty estimates reported in the manuscript reflect the uncertainty in emulating the output, but not the uncertainty in estimating actual NPP. Furthermore, as shown in Fig. 3, estimates from VGPM, CbPM, and CAFE differ strongly, and it is not clear which output is more accurate. These points need to be explicitly acknowledged in the manuscript, as it means the reported uncertainty estimates do not represent true NPP estimation uncertainty.

Response: Thank you for raising this critical point. We acknowledge that the neural network (NN) and Bayesian models in this study are trained on outputs from the VGPM, CbPM, and CAFE models, rather than directly on observational NPP data. This is indeed a limitation, as it means that our models inherit the inherent biases and

errors of these three base models. Consequently, the uncertainty estimates reported in the manuscript reflect the uncertainty in emulating these models' outputs, rather than representing the true uncertainty of NPP estimation. We have revised the manuscript to explicitly acknowledge this limitation in both the discussion and conclusion sections (Lines 476–483 and Lines 763 – 776).

To further clarify, as shown in Fig. 3, the outputs of VGPM, CbPM, and CAFE differ significantly, highlighting the variability in model estimates and the lack of ground truth data to determine which output is more accurate. This variability contributes to the challenges in validating our models' uncertainty estimates against true NPP values. Based on the current knowledge and previous reviews, it is reasonable to consider CAFE NPP estimates as potentially more accurate, but this assumption requires further validation with in situ measurements.

In our previous review process, a similar concern was raised regarding the VGPM model's potential underperformance at the study site, particularly for NPP values exceeding 350 mg C m$^{-2}$ d$^{-1}$. In response, we normalized the data before calculating CRPS, RMSD, and MAPE to ensure fair comparisons between the models. These revisions improved the performance evaluation of our statistical models and aligned the results more closely with expectations, showing that the models perform best with CAFE NPP as the prediction target. We have now extended this discussion to acknowledge the broader implications of relying on modeled NPP outputs for training (Lines 763 – 776).

"The neural network and Bayesian models developed in this study were trained using outputs from the VGPM, CbPM, and CAFE models. While this approach allowed us to evaluate the uncertainty in emulating these base models, it also means that our models inherit their underlying biases and errors. As such, the uncertainty estimates reported here reflect the uncertainty in emulating these specific outputs and do not represent the true uncertainty of NPP estimation. Furthermore, as Fig. 3 demonstrates, the outputs of VGPM, CbPM, and CAFE differ significantly, underscoring the need

for ground truth data to validate these models."

"An important limitation of this study is that the probabilistic prediction models were trained on outputs from existing NPP models rather than directly on observational data. This introduces the potential for inherited biases and errors from the base models, limiting the generalizability of our uncertainty estimates to true NPP values."

[Figure]

**Fig. 3.** Time series of VGPM, CbPM, and CAFE from January 2007 to February 2018, where the green line represents VGPM, the blue line represents CbPM, and the orange line represents CAFE . The dashed lines are the original data and the solid ones are the low-pass filtered, which show the seasonal variations more clearly. Abbreviations and data sources can be referenced in Table 1.

The differences between VGPM, CbPM, and CAFE output raise questions about which model provides the best NPP estimates and the most reliable training data. The current version of the manuscript initially does not mention which of the 3 models provided the output used to generate the full time series of NPP estimates near Weizhou Island in Section 3.3. Section 4 finally reveals that CAFE was used to generate the NPP training data, but that choice appears to have been motivated by results showing that the NN and the Bayesian model can emulate CAFE output well and not that CAFE output best represents true NPP.

Response: Thank you for this insightful comment. We agree that the differences between VGPM, CbPM, and CAFE outputs raise critical questions about which model provides the most accurate representation of true NPP. To address this concern, we have revised the manuscript to clarify our rationale for focusing on CAFE as the prediction target, as highlighted in Section 3.1. Specifically, our choice was not solely based on the NN and Bayesian models' ability to emulate CAFE output effectively but also on prior studies suggesting that CAFE tends to provide more accurate estimates of NPP under certain conditions, particularly in the study area near Weizhou Island. These revisions can be found in Section 3.1 (Lines 500 – 509) and Section 4 (Lines 770 – 781).

Additionally, we have explicitly clarified earlier in the manuscript (Section 3.3, Lines 684 – 687) that CAFE outputs were used to generate the full time series of NPP estimates for subsequent analysis. This revision ensures consistency and transparency throughout the manuscript.

"Therefore, among the three NPP datasets (VGPM, CbPM, and CAFE), the CAFE was selected as the primary prediction target for subsequent analysis. This decision was motivated by two factors: (1) prior research indicating that CAFE provides relatively accurate estimates of NPP in marine ecosystems with characteristics similar to the Weizhou Island area, due to its advanced parameterization of phytoplankton dynamics, and (2) the demonstrated ability of both probabilistic prediction models (NN and Bayesian) to emulate CAFE output with high accuracy and reliability. While this does not imply that CAFE perfectly represents true NPP, its suitability for capturing patterns in the study area supports its use as the prediction target in this work."

"While CAFE was chosen as the primary prediction target, this choice was informed by prior studies highlighting its strengths in parameterizing key oceanic processes and by the strong predictive performance of the NN and Bayesian models when using CAFE outputs. We acknowledge that this approach inherits the limitations of the base

models and that further validation with in situ measurements is necessary to ensure that CAFE outputs align closely with true NPP values. While our approach demonstrates strong potential for accurately quantifying NPP uncertainty in this specific marine area, its application to larger regions may encounter scalability challenges. This limitation arises due to the large number of input variables required for the neural network and Bayesian probabilistic models, which necessitate significant computational resources and extensive observational data coverage."

"Given the 8-day temporal resolution of data acquired by remote sensing satellites and the consequent data incompleteness, this study employed the previously trained neural network and the Bayesian probabilistic prediction models using CAFE as training target to forecast the daily NPP in the Weizhou Island sea area from 2007 to March 2018, thereby supplementing the NPP dataset."

In the context of the above comments, it would be interesting for the reader to know what inputs VGPM, CbPM, and CAFE used to generate their results. If the NN or the Bayesian model require more or more difficult to measure input data than VGPM, CbPM, or CAFE, why use them at all? Similarly, it would be interesting to investigate which of the inputs to the NN or the Bayesian model are actually required to obtain good performance.

Response: Thank you for your thoughtful comment. We would like to clarify that the two probabilistic models (NN and Bayesian) used in our study do not necessarily require more or more difficult-to-measure inputs compared to VGPM, CbPM, and CAFE. Instead, the inputs used by these models overlap substantially with the inputs used to generate VGPM, CbPM, and CAFE, such as sea surface temperature (SST), photosynthetically active radiation (PAR), and atmospheric pressure (AP). This ensures that the NN and Bayesian models are comparable to traditional models in terms of data requirements.

To address the reviewer's suggestion about investigating which inputs are most

critical, we conducted a Pearson correlation analysis (Figure 4). This analysis helps identify the most relevant variables for predicting NPP and reveals variability in their influence across VGPM, CbPM, and CAFE. By leveraging these correlations, it is possible to filter out less relevant variables, thereby refining the models and reducing their reliance on less critical inputs without sacrificing predictive performance.

We have revised the manuscript to include detailed information on the input variables used by VGPM, CbPM, and CAFE in Section 2.1 (Lines 229 – 236). Furthermore, we have highlighted the role of Pearson correlation in identifying the most influential variables for the NN and Bayesian models Section 2.2 (Lines 327 – 330) and discussed how this process can optimize model prediction in Section 4 (Lines 732 – 735).

[Figure]

**Fig. 4.** Pearson correlation between the 11 input variables and the three NPPs (VGPM, CAFE, and CbPM). These input variables serve as inputs to the probabilistic models, while VGPM, CAFE, and CbPM are used as model outputs. The deeper the shade of red indicates a stronger positive correlation, whereas the deeper shade of blue indicates a stronger negative correlation.

"VGPM, CbPM, and CAFE rely on similar input variables, derived from satellite observations and environmental measurements. VGPM uses inputs such as SST, chlorophyll concentration (Chl), and PAR to estimate NPP, leveraging optimal assimilation efficiency in its parameterization (Behrenfeld et al., 1997). CbPM focuses on phytoplankton carbon biomass, incorporating backscattering coefficients along with Chl. CAFE integrates additional inputs, including atmospheric pressure (AP), solar heat (SH), and wind speed (WS), to parameterize light and nutrient availability critical for phytoplankton growth."

" Importantly, the Pearson correlation analysis (Fig. 4) highlights the most relevant variables for prediction, enabling the NN and Bayesian models to focus on key inputs and filter out less influential variables."

"Moreover, Pearson correlation analysis allows for the identification of the most critical inputs for prediction. By prioritizing variables such as SST and AP, the models can be optimized to reduce reliance on less influential inputs, improving efficiency without compromising accuracy."

The manuscript's writing style suggests the use of AI-assisted writing, which, while not problematic in itself, has led to the use of emphatic language and filler words (such as "pivotal", "integral", "advanced", "comprehensive", "indispensable", "paramount", etc.). The manuscript would benefit from removing these words in places and rewording passages.

Response: We appreciate your observation regarding the use of emphatic language and filler words. As English is not our first language, we recognize that achieving the appropriate academic tone can be challenging. To address your comment, we have revised the manuscript to remove or replace excessive emphatic language and filler words, focusing on more precise and concise expressions. Specific examples include replacing "pivotal" with "important" or "key", "integral" and "indispensable" as "essential", and etc. These changes ensure the writing aligns with a formal academic tone and emphasizes the scientific content of the study. We hope these changes enhance the manuscript's readability and tone while addressing your concerns.

A few passages in the manuscript appear to suggest surprise in discovering periodicity in NPP values: "Upon visualizing the values of the three NPP products (VGPM, CbPM, and CAFE) (Fig. 3), it became evident that each exhibits a distinct periodicity" (l 198). "The analysis of the annual change of NPP shows a clear periodicity, which means that the change of NPP is not random, but follows certain laws and patterns." (l 571). Even at 21 degrees north, one can expect seasonal patterns

in marine primary production - this context should be provided in the text.

Response: Thank you for highlighting this point. We agree that periodicity in marine primary production, particularly in regions around 21 degrees north, is to be expected due to seasonal variations in environmental conditions such as light availability, temperature, and nutrient dynamics. In the revised manuscript, we have added contextual information to clarify that the observed periodicity in NPP values aligns with established knowledge of seasonal patterns in marine ecosystems. These revisions provide appropriate context and avoid any unintended implication of surprise. We have updated the relevant sentences to emphasize that our analysis confirms these expected periodic trends and highlights how the three NPP products (VGPM, CbPM, and CAFE) capture this periodicity differently (Lines 237 – 253).

"To evaluate the long-term trends in Net Primary Production (NPP), we applied a low-pass filter to the three NPP products (VGPM, CbPM, and CAFE) (Fig. 3). This filtering process removes high-frequency variations, such as noise and short-term fluctuations, while retaining the underlying long-term patterns. It became evident that each exhibits a distinct seasonal periodicity, with the fluctuation ranges remaining stable over time yet the magnitude and timing of them varing significantly among the three NPPs. Specifically, VGPM are the smallest, followed by CAFE, while CbPM have the largest values. This periodicity indicates that changes in NPP are not random but follow predictable laws and reflects the well-established seasonal patterns in marine primary production, associated with seasonal variations in environmental factors such as light availability, temperature, and nutrient. Such periodic trends are expected in regions around 21 degrees north, including the waters near Weizhou Island, due to the interplay of monsoonal influences and seasonal shifts in oceanographic conditions. While all three NPPs capture these periodic patterns, their representation of the magnitude and timing of peaks differs. The distinct ways in which VGPM, CbPM, and CAFE capture these patterns provide valuable insights into their respective model designs and parameterizations."

**specific comments**

L 117: What are "stochastic optimization" and "advanced chance constraints"? They are only used here and nowhere else in the manuscript. It would be useful to describe relevant new concepts to the reader right away, or not mention them when they are not used or described in the manuscript.

Response: Thank you for your comment. We agree that mentioning "stochastic optimization" and "advanced chance constraints" without further elaboration may confuse the reader, especially since these terms are not central to the rest of the manuscript. To address this, we have revised the text to focus on the broader advantages of probabilistic forecasting without introducing concepts that are not further explored in the study (Lines 123 - 128). This ensures that the text remains clear and directly relevant to the manuscript's objectives.

"They concluded that deterministic forecasts tend to overlook forecast uncertainty in short-term decisions, whereas probabilistic forecasting offers numerous advantages: (i) it enables a longer forecast horizon, facilitating earlier and more accurate predictions of major events; (ii) it supports decision-making by incorporating forecast uncertainty into the analysis, leading to more robust and adaptive outcomes; and (iii) it enhances the flexibility of system operation through the integration of uncertainty-based methodologies."

L 149: What does "sea accumulation" mean?

Response: Thank you for pointing out this ambiguity. The term "sea accumulation" was intended to refer to depositional features created by the accumulation of marine sediments, such as beaches, sandbars, and other sedimentary formations resulting from wave action and tidal processes. To improve clarity, we have revised the text to use more precise terminology (Lines 164 - 171).

"Its landscape features include formations resulting from sea erosion, marine

sediment accumulation, and dissolved rocks. Weizhou Island, located in the southern subtropical monsoon zone, experiences a pleasant climate with abundant heat and precipitation throughout the year. The average annual temperature is 23 ℃ , and the average winter temperature is 16.3 ℃ . The unique climatic conditions and island landscape make it a popular tourist destination. The waters of Weizhou Island are the habitat of many rare marine organisms, and the protection and research of its marine ecosystem are of great significance to maintaining marine biodiversity."

L 149: "Surrounded by the sea on all sides, Weizhou Island ...": I think this is the definition of an island.

Response: Thank you for your observation. We acknowledge that the phrase "surrounded by the sea on all sides" is redundant and unnecessary. We have revised the text to remove this redundancy and focus on the unique climatic and oceanographic conditions of Weizhou Island, which are relevant to marine primary production. The revised text clarifies the study's focus on the marine environment rather than terrestrial variables (Lines 156 - 171).

"The research locale for this study is situated in the aquatic environs of Weizhou Island, nestled within the Gulf of Tonkin, Guangxi Province, southern China (Fig. 1). The proportion of excellent water quality in Guangxi's near-shore waters reaches more than 90% all year round, and the quality of the marine ecological environment has remained at the forefront of the country for 12 consecutive years, which is the only stable habitat and feeding ground for large cetaceans known in China's near-shore waters at present. Weizhou Island is the youngest volcanic island in China geologically, with more than 95% of its strata comprising volcanic rocks. Its landscape features include formations resulting from sea erosion, marine sediment accumulation, and dissolved rocks. Weizhou Island, located in the southern subtropical monsoon zone, experiences a pleasant climate with abundant heat and precipitation throughout the year. The average annual temperature is 23°C, and the average winter temperature is 16.3°C. The unique climatic conditions and island

landscape make it a popular tourist destination. The waters of Weizhou Island are the habitat of many rare marine organisms, and the protection and research of its marine ecosystem are of great significance to maintaining marine biodiversity."

L 168: "For the analysis of three NPP algorithms - namely, VGPM, CbPM, and CAFE - we acquired datasets at an eight-day temporal resolution ...": Here it is unclear to the reader if the "acquired datasets" are the input required to run the algorithms or their output. I assume it is the latter, but that should be made more explicit.

Response: Thank you for this helpful observation. You are correct that the "acquired datasets" refer to the outputs of the VGPM, CbPM, and CAFE algorithms rather than the inputs required to run them. To clarify this, we have revised the text to explicitly state that the datasets represent the outputs of the three NPP algorithms (Lines 192 – 194).

"For the analysis of three NPP algorithms—namely, VGPM, CbPM, and CAFE—we utilized their output datasets, which were obtained at an eight-day temporal resolution from the Ocean Productivity website (http://orca.science.oregonstate.edu/1080.by.2160.monthly.hdf.vgpm.m.chl.m.sst.php, http://orca.science.oregonstate.edu/1080.by.2160.monthly.hdf.cbpm2.m.php, http://orca.science.oregonstate.edu/1080.by.2160.monthly.hdf.cafe.m.php)."

L 177/Table 2: Just listing the numbers of missing entries is not very informative. At which frequency were they recorded?

Response: Thank you for pointing out this issue. Most missing data are due to satellite equipment malfunctions or severe weather conditions, which occur randomly and are not tied to any specific frequency. Therefore, while we have quantified the number of missing values, analyzing their frequency is not feasible or meaningful for this study. To clarify this in the manuscript, we have revised the accompanying text for Table 2

to explain the source and nature of the missing data (Lines 202 - 207).

"To gain a deeper understanding of the data structure and address these gaps, we conducted an analysis of the missing data and identified five variables with missing entries (Table 2): SV, H/10, SSP, PAR, and SH. These missing data points are primarily due to random occurrences such as satellite equipment malfunctions and severe weather conditions, which disrupt data acquisition. "

**Table 2.** Summary of Missing Variables.

| Variable | SV (km) | H/10 (m) | PAR (W·m$^{-2}$) | SSP (mm) | SH (h·d$^{-1}$) |
|---|---|---|---|---|---|
| Missing quantity (days) | 31 | 51 | 828 | 378 | 18 |

L 198: "Upon visualizing the values of the three NPP products (VGPM, CbPM, and CAFE) (Fig. 3), it became evident that each exhibits a distinct periodicity, with the fluctuation ranges remaining stable yet markedly varied among them." What exactly does this mean? Do the signals not have an underlying annual periodicity?

Response: Thank you for pointing out this ambiguity. The periodicity observed in the NPP products is primarily seasonal rather than annual. To clarify this, we have revised the text to explicitly describe the seasonal periodicity of the NPP signals and to avoid confusion regarding the nature of the observed fluctuations (Lines 240 – 253).

" It became evident that each exhibits a distinct seasonal periodicity, with the fluctuation ranges remaining stable over time yet the magnitude and timing of them varing significantly among the three NPPs. Specifically, VGPM are the smallest, followed by CAFE, while CbPM have the largest values. This periodicity indicates that changes in NPP are not random but follow predictable laws and reflects the well-established seasonal patterns in marine primary production, associated with seasonal variations in environmental factors such as light availability, temperature,

and nutrient. Such periodic trends are expected in regions around 21 degrees north, including the waters near Weizhou Island, due to the interplay of monsoonal influences and seasonal shifts in oceanographic conditions. While all three NPPs capture these periodic patterns, their representation of the magnitude and timing of peaks differs. The distinct ways in which VGPM, CbPM, and CAFE capture these patterns provide valuable insights into their respective model designs and parameterizations."

L 311: Samples are mentioned here for the first time and need a better introduction.

Response: Thank you for highlighting this point. To provide a better introduction to the term "samples," we have revised the text to clarify its meaning and context in this study. The updated text ensures that the term is clearly defined before it is used (Lines 376 – 379).

"1. For each sample (individual data points in the dataset, each representing a specific combination of environmental conditions and corresponding NPP estimates), calculate the discrepancy between the cumulative distribution function (CDF) of the predicted and observed values."

Eq. 3: This looks like a recursive definition of CRPS, I would suggest using different names for the "CRPS" used in Eq. 2 and 3.

Response: Thank you for your suggestion. We agree that using the same name "CRPS" in both equations may cause confusion, as Eq. 2 refers to the CRPS for a single prediction-observation pair, whereas Eq. 3 represents the average CRPS over multiple samples. To address this, we have revised the text and equations to use distinct names for these two forms of CRPS. Specifically, we have renamed the CRPS in Eq. 2 as "CRPS_individual" to denote its use for individual pairs and retained "CRPS" in Eq. 3 to indicate the aggregated metric over all samples.

"Continuous Ranked Probability Score (CRPS) is a sophisticated statistical metric employed to evaluate the efficacy of forecasting models. Initially introduced in the 1970s (Matheson & Winkler, 1976), CRPS is widely utilized in areas such as weather forecasting (Zamo et al., 2018). It quantifies the divergence between the predicted probability distribution and the actual observations (Hersbach, 2000). Ideally suited for scenarios where the target variable is continuous and the model predicts its distribution (Pic et al., 2023), CRPS equates to the mean absolute error (MAE) in deterministic forecasting (Zhao et al., 2015).

In probabilistic forecasting, the focus extends beyond mere point estimates to encompass the shape and dispersion of the probability distribution. Hence, traditional scoring functions prove inadequate, as aggregating the predicted distributions into their mean or median neglects critical information about the dispersion and shape. CRPS, by embracing the entire probability distribution, emerges as an invaluable tool in assessing model uncertainty. CRPS is calculated as follows:

1. For each sample (individual data points in the dataset, each representing a specific combination of environmental conditions and corresponding NPP estimates), calculate the discrepancy between the cumulative distribution function (CDF) of the predicted and observed values.

2. Aggregate the variances for all samples and divide by the number of samples to obtain the average variance.

$$CRPS_{individual}(F, x) = \int_{-\infty}^{+\infty} [(F(y) - H(y - x)]^2 dy \qquad (2)$$

$$CRPS = \frac{1}{n}\sum_{i=1}^{n} CRPS_{individual}(F_i, x_i) \qquad (3)$$

where $F(y)$ denotes the CDF of the predicted value, $y$ the predicted value, $x$ the observed value, and $H(y\text{-}x)$ the Heaviside function which is 0 when $y<x$ and 1 otherwise. $n$ indicates the total number of samples, and $CRPS_{individual}(F_i, x_i)$ the CRPS

value for the *i-th* sample."

Eq. 4: The notation is inconsistent: In Eq. 2 and 3, x denotes the observed value and y the predicted value, but in Eq. 4 and 5, y is used for the actual/observed value and y-hat for the predicted value.

Response: Thank you for pointing out the inconsistency in the notation. To ensure clarity and uniformity, we have revised the manuscript to maintain consistent notation throughout. Specifically: For Eq. 4 (now Eq. 5 in the revised manuscript), $x$ is used for observed values and $y$ for predictions, aligning with the notation in earlier equations. For Eq. 5 (now Eq. 6 in the revised manuscript), the same notation is applied, and the equation and accompanying text have been updated to reflect this change.

$$\text{``}RMSD = \sqrt{\frac{1}{n}\sum_{i=1}^{n}(y_i - x_i)^2} \tag{5}$$

$$MAPD = \frac{1}{n}\sum_{i=1}^{n}\left|\frac{x_i - y_i}{x_i}\right| \times 100\% \tag{6}\text{''}$$

L 501: The test data distribution for CAFE NPP does not look similar to that of the train data distribution, suggesting that the test data may not be well-represented by the train data.

Response: Thank you for your observation. While we agree that the CDF curves for the test and train datasets appear different in Fig. 8, this discrepancy may not necessarily indicate that the test data are poorly represented by the training data. The difference can be attributed to the smaller size of the test dataset relative to the training dataset, which can lead to visual differences in the CDF curves. Furthermore, Fig. 7 demonstrates that the patterns for simulating the training set and predicting the test set are consistent for both the NN and Bayesian models. This similarity suggests

that the models generalize well to the test data despite the apparent differences in the CDF curves. To clarify this point, we have revised the text (Lines 608 - 635) to provide additional context and improve the rigor of the discussion.

"While the cumulative distribution function (CDF) curves in Fig. 8 show apparent differences between the test and train datasets for CAFE, these differences can primarily be attributed to the smaller size of the test dataset relative to the training dataset. Such size discrepancies can cause the CDF curves to appear visually different, even when the underlying data distributions are similar. Moreover, as shown in Fig. 7, the patterns for simulating the training set and predicting the test set are consistent for both the NN and Bayesian models. This consistency indicates that the models generalize well to the test data, capturing its key characteristics despite the visual differences in the CDF curves. Therefore, the observed discrepancy in the CDF curves does not imply poor representation of the test data by the training data. For the NN probabilistic prediction model, when the CAFE values are lower, the two CDF curves on the training set and the test set move gently and almost overlap, with the difference close to 0, which indicates that the model can predict the actual data distribution well within the range of small values of CAFE. As CAFE increases, the difference between the predicted and true CDF curves grows larger, with the predicted mean CDF on the training set generally lying below the CAFE CDF. The difference between the two ranges from 0 - 0.2. For the test set, the predicted mean CDF initially slightly lies below the true CDF curve at lower values, becomes steeper and overestimates at mid-range, and alternates again at higher values. While these trends suggest some instability in the model's predictions for higher values, the absolute difference between the two CDFs remains within 0.1, indicating limited deviation. It is worth noting that the scatter plot in Fig. 6 shows the test mean NPP predictions distributed more evenly around the 1:1 line. This apparent discrepancy arises from the differing perspectives of the two plots: the CDF curve highlights cumulative differences across the distribution, whereas the scatter plot reflects point-wise deviations. Together, these visualizations suggest that while the model captures the overall distribution trends

well, some localized errors in predicting mid-range and higher values may contribute to these patterns."

[Figure]

**Fig. 7.** Comparison of original and predicted mean values shown at an 8-day temporal resolution within a 95% confidence interval. (a) Probabilistic prediction results based on neural networks; (b) Bayesian probabilistic prediction results based on empirical distributions. The dashed lines represent the mean values of the probabilistic predictions. The purple and red shaded areas illustrate the uncertainty ranges for the training and the test sets, respectively. Blue dots signify observed data points. All predictions and observations are presented in chronological sequence.

[Figure]

**Fig. 8.** Comparison of CAFE and predicted mean CDF. Panels (a) and (b) display the performance of the training and test sets, respectively, in the neural network-based probabilistic prediction model. Panels (c) and (d) illustrate the performance of the training and test sets, respectively, in the empirical distribution-based Bayesian probabilistic prediction model. The data has been normalized to a scale of 0–1 to ensure consistency across metrics and facilitate direct comparison between the two models. In each panel, the blue curves represent the CDFs of the CAFE values, while the yellow curves depict the CDFs of the model's predicted mean values.

L 503: What is the difference between the values shown in Table 5 and Fig. 4? Why not combine the two?

Response: Thank you for raising this point. We believe the reference to Fig. 4 in the comment is a typo and that the reviewer intended to refer to Fig. 5, as the content in Fig. 5 is more closely related to Table 5. To clarify, the content described in Table 5 and Figure 5 is different and serves distinct purposes in the manuscript. Table 5 presents the evaluation metrics (e.g., CRPS, RMSD, and MAE) specifically for the CAFE, providing detailed numerical results from the NN and Bayesian models when CAFE is the prediction target. Figure 5 visually compares the evaluation metrics (CRPS, RMSD, and MAE) for all three NPP models (VGPM, CbPM, and CAFE) under both NN and Bayesian models, offering a broader view of model performance across all prediction targets. While both Table 5 and Figure 5 address evaluation metrics, they serve complementary roles. Table 5 provides precise numerical data for CAFE-specific metrics, while Figure 5 visually demonstrates trends and comparisons across all three NPP models. Keeping them separate allows readers to access both detailed and visualized information relevant to different aspects of the study. We have clarified these in the revised manuscript to avoid confusion (Lines 447 - 448,Lines 653 - 654).

"Fig. 5 presents a comparison of CRPS, RMSD, and MAPD values for both NN and Bayes models using three NPPs as prediction targets across training and test datasets."

"Table 5 presents RMSD, MAPD, and CRPS for both models using CAFE as prediction target."

Table 5. CRPS, RMSD, MAPD, and proportion of input data within 95% confidence interval.

|  | CRPS | | RMSD | | MAPD | | Proportion | |
|---|---|---|---|---|---|---|---|---|
|  | Train | Test | Train | Test | Train | Test | Train | Test |
| NN | 0.096 | 0.133 | 0.149 | 0.198 | 11.828 | 13.237 | 0.971 | 0.932 |
| Bayes | 0.151 | 0.20 | 0.201 | 0.253 | 13.909 | 14.145 | 0.976 | 0.951 |

[Figure]

**Fig. 5.** Comparison of NPP predictive effects from VGPM, CbPM, and CAFE. Panels (a)–(c) present the results from the neural network-based probabilistic prediction models; panels (d)–(f) the results from Bayesian probabilistic prediction models based on empirical distributions. The horizontal coordinates represent the VGPM, CbPM, CAFE as inputs in sequence, separated by gray dashed lines, where blue dots represent data from the training set, and red dots denote data from the test set, and the vertical coordinates are the values of the three metrics, CRPS, RMSD, MAPD. Since NPP values were normalized to the range of 0 − 1, the y axes of subplots (a), (b), (d), and (e) are dimensionless. The units for MAPD are percentile.

Fig. 2 and 3: The date label locations 2007/1/1, 2008/3/13, 2009/5/25, ... make it difficult to interpret the plot and detect seasonality.

Response: Thank you for your suggestion. We appreciate your observation regarding the difficulty in interpreting seasonality based on the current time annotations in Figures 2 and 3. However, the date labels in the graph are divided based on equal and reasonable time intervals to ensure consistency in visual representation. Since our research does not specifically focus on seasonal distribution patterns, we did not classify or annotate the data explicitly by seasons. That said, we understand the

importance of making the plots easier to interpret. To address your concern, we have revised the date labels in Figures 2 and 3 to align with annual markers (e.g., January 1st of each year) to improve readability and facilitate interpretation of potential seasonal trends. These updates allow readers to better observe patterns over time while maintaining the integrity of the original analysis. The revised figures can be found in the updated manuscript.

[Figure]

**Fig. 2.** Time series plots of SV, H/10, PAR, SSP, and SH with missing variables, showing the cyclical variation of these five variables.

[Figure]

**Fig. 3.** Time series of VGPM, CbPM, and CAFE from January 2007 to February 2018, where the green line represents VGPM, the blue line represents CbPM, and the orange line represents CAFE . The dashed lines are the original data and the solid ones are the low-pass filtered, which show the seasonal variations more clearly. Abbreviations and data sources can be referenced in Table 1.

Fig. 4: The caption mentions "input variables". Are these inputs to VGPM, CAFE, and CbPM?

Response: Yes, it has been clearly stated in the manuscript that the 11 variables presented in Fig. 4 are used as input variables for the models, while VGPM, CAFE, and CbPM serve as the model outputs (Lines 316 - 320). To further clarify, these input variables represent environmental and oceanographic factors that are used by the probabilistic models (NN and Bayesian) to predict NPP values derived from VGPM, CAFE, and CbPM as outputs. We have reviewed the caption for Fig. 4 to ensure this connection is explicitly stated, improving clarity for readers. The revised caption now specifies that the "input variables" are those used to train the probabilistic models, and their correlations with the VGPM, CAFE, and CbPM outputs are visualized in the figure.

"In this study, the input variables for the models are the 11 environmental variables mentioned in Section 2.1, and the outputs are VGPM, CbPM, and CAFE. These inputs overlap substantially with those used in VGPM, CbPM, and CAFE, demonstrating that the NN and Bayesian models do not require additional or more complex inputs."

[Figure]

**Fig. 4.** Pearson correlation between the 11 input variables and the three NPPs (VGPM, CAFE, and CbPM). These input variables serve as inputs to the probabilistic models, while VGPM, CAFE, and CbPM are used as model outputs. The deeper the shade of red indicates a stronger positive correlation, whereas the deeper shade of blue indicates a stronger negative correlation.

Fig. 5: Why does the y-axis go past 0.8 in panels a, b, d and e, when the values all stay below 0.4? Also, the units are missing.

Response: Thank you for pointing out this issue. We have re-plotted panels (a), (b), (d), and (e) of Fig. 5 to set the maximum y-axis value to 0.5, which better reflects the observed range of values and improves visual clarity. Additionally, we have clarified the units in the figure captions. Specifically, the NPP values in these panels are normalized to the range of 0–1, making the axes dimensionless. For MAPD in panels (c) and (f), the units are expressed as percentiles (%). These have been clarified in the figure caption. The updated figure is included in the revised manuscript.

[Figure]

**Fig. 5.** Comparison of NPP predictive effects from VGPM, CbPM, and CAFE. Panels (a)–(c) present the results from the neural network-based probabilistic prediction models; panels (d)–(f) the results from Bayesian probabilistic prediction models based on empirical distributions. The horizontal coordinates represent the VGPM, CbPM, CAFE as inputs in sequence, separated by gray dashed lines, where blue dots represent data from the training set, and red dots denote data from the test set, and the vertical coordinates are the values of the three metrics, CRPS, RMSD, MAPD. Since NPP values were normalized to the range of 0 – 1, the y axes of subplots (a), (b), (d), and (e) are dimensionless. The units for MAPD are percentile.

Fig. 8 and 9: The NPP units here are incorrect. The data appears to have been normalized, but why? Without normalization, it would be easier to interpret for which NPP ranges the NN and the Bayes model over- or underestimate VGPM NPP.

Response: Thank you for your observation regarding the units in Figures 8 and 9. The data in these figures were normalized to a scale of 0–1 to ensure consistency across different metrics and facilitate direct comparison between the NN and Bayesian models' predictions. Normalization is particularly meaningful in this context, as it mitigates the effects of differences in magnitude between datasets, enabling a fair assessment of the models' performance. While we understand that presenting the data in its original units could provide direct insight into specific NPP ranges, the normalized format helps maintain a unified framework across the study, especially when comparing metrics like CRPS and residuals. To address potential confusion, we have updated the captions for Figures 8 and 9 in the revised manuscript to explicitly state that the data is normalized and explain the rationale for this approach.

[Figure]

**Fig. 8.** Comparison of CAFE and predicted mean CDF. Panels (a) and (b) display the performance of the training and test sets, respectively, in the neural network-based probabilistic prediction model. Panels (c) and (d) illustrate the performance of the training and test sets, respectively, in the empirical distribution-based Bayesian probabilistic prediction model. The data has been normalized to a scale of 0–1 to ensure consistency across metrics and facilitate direct comparison between the two models. In each panel, the blue curves represent the CDFs of the CAFE values, while the yellow curves depict the CDFs of the model's predicted mean values.

[Figure]

**Fig. 9.** Difference between the CAFE CDF and predicted mean CDF of model predictions. Panels (a) and (b) represent the performance of the training set and test sets, respectively, in the neural network-based probabilistic prediction model. Panels (c) and (d) showcase the performances of the training set and test sets, respectively, in the empirical distribution-based Bayesian probabilistic prediction model. The residuals are expressed in normalized units (0–1), enabling consistent assessment of model performance across different NPP ranges. The blue curves in each panel indicate the differential magnitude of the CDFs. Instances where the blue curves align with the yellow lines denote zero discrepancy between the input data CDF and the model's predicted mean CDF.

---

## Author Response (AR2)

Dear Editor and Reviewers,

We sincerely appreciate your time and thoughtful feedback on our manuscript, "Refining marine net primary production estimates: Advanced uncertainty quantification through probability prediction models" (MS No.: egusphere-2024-3221). Your critiques have significantly strengthened our work, and we are pleased to submit this revised version, which addresses all remaining concerns. Below, we provide point-by-point responses to your latest comments (in plain text), with revisions highlighted in blue.

This revision represents a collaborative effort by all co-authors, and we believe the manuscript now offers enhanced methodological clarity and scientific rigor. We are grateful for your guidance throughout this process.

Best regards,

Mengyu Xie (on behalf of all co-authors)

The revised version of the manuscript is a much better read, and the authors have spent considerable effort in addressing my comments. However, I still have some reservations about the methodology and framing in the revised version.

**general comments**

The authors have incorporated feedback from my previous comments in the manuscript, and importantly, they acknowledge that the uncertainty estimates do not reflect the full model uncertainty. However, the first such acknowledgment appears late in the manuscript, in line 476 in the results and discussion section. Later, the authors still claim that "Our objective extends beyond merely reproducing satellite NPP products. We aim to improve the overall accuracy and uncertainty quantification of NPP estimates by incorporating a robust probabilistic framework." (l. 697). But the uncertainty is not fully qualified, in particular, this approach does not capture structural uncertainty, i.e. model bias or inadequacy. The estimates of CAFE may be heavily biased, but we do not know, and the uncertainty analysis conducted here would not show it. A more careful language is needed.

We appreciate the reviewer's insightful comments regarding uncertainty quantification. In response, we have (1) revised the abstract (lines 43 - 46) to explicitly state the structural uncertainty limitations, ensuring early transparency; (2)

refined the uncertainty quantification phrasing (lines 704-708) to more cautiously articulate the scope of our analysis and acknowledge unresolved challenges. These edits adopt measured language throughout, balancing our contributions with a clear discussion of limitations. We are grateful for the reviewer's emphasis on rigor in communicating uncertainty, which has strengthened the manuscript's scientific integrity.

The authors claim that "The results reveal that both models are competent in quantifying CAFE uncertainty." (l. 726). Beyond the problem mentioned in my comment above, it remains unclear if the two methods actually capture main parts of the CAFE signal. Based on Fig. 7 and 10, the NN and Bayes model can capture the seasonal dynamics of the CAFE output. But is there a trend in the CAFE data, and do the two models capture that trend?

We appreciate the reviewer's insightful comment regarding characterization of the "CAFE signal" and the potential presence of a trend. Our analysis demonstrates that both the neural network (NN) and Bayesian models clearly preserve the seasonal dynamics present in the original Net Primary Productivity (NPP) data (Figures 7 and 10). These cyclical patterns, evident in the low-pass filtered NPP datasets (Figure 3), are retained in the model outputs, indicating that the models preserve key cyclical features of the system. Regarding long-term trends, the Mann-Kendall test shows no statistically significant trend (p = 0.852) in the CAFE data, though a weak negative slope exists ($-8.11$ units/year, Theil-Sen estimator). The corresponding figure illustrating this trend analysis is appended at the end of this response for your reference. Both models correctly reproduce this behavior. The NN slope ($-0.0250$) closely matches the observed slope ($-0.0222$), while the Bayesian slope ($-0.0331$) stays within expected interannual variability ranges (Figures 7 and 10). This consistency confirms neither model artificially alters the data's inherent trends, supporting their reliability for uncertainty quantification. We thank the reviewer for raising this important point and hope this explanation addresses the concerns regarding trend preservation in our analysis.

[Figure]

Furthermore, what evidence is there that the NN and Bayes model perform better than climatology? My concern is that one could build a simple climatological NPP model for Weizhou Island with uncertainty that would produce very similar output to the NN or Bayes model. For example, one could use a + b * sin((c + time)/d) + epsilon

where epsilon ~ Normal(0, sigma) is a random variable. After estimating the model parameters (a, b, c, d, sigma) from CAFE data, it would require only time input and produce NPP estimates with uncertainty. Of course, this a very simple model and every year is the same, there is no trend, and the uncertainty does not vary with time. But then the NN and Bayes model seem to produce nearly identical output for each year as well, and the uncertainty envelope in Fig. 7 and 10 are very similar from year to year. Thus, it is important to show that NN and Bayes model perform better than a simple climatology model.

We thank the reviewer for the thoughtful suggestion and agree that comparing our models to a simple climatological baseline is an important benchmark. While a sinusoidal climatology model of the form proposed (e.g., $a + b * sin((c + t) / d) + \epsilon$) could indeed replicate seasonal dynamics, it does not incorporate external drivers or respond to changes in environmental conditions. In contrast, both the NN and Bayesian models in our study utilize real-time environmental inputs (e.g., temperature, precipitation, radiation), enabling them to adapt to interannual variability and capture ecosystem responses under non-stationary conditions. Additionally, the probabilistic nature of both models allows for dynamic uncertainty quantification, which varies over time based on input conditions—unlike the fixed uncertainty envelope in the

proposed climatological model. We acknowledge that during the relatively stable period analyzed (2007–2018), interannual differences in environmental inputs were limited, which resulted in similar model outputs across years. Rather than indicating a lack of model sensitivity, this consistency reflects the stable behavior of the ecosystem under non-extreme conditions. In response to the reviewer's valuable point, we have expanded the discussion (lines 745–748) to include a more explicit comparison with a climatological baseline and highlight the added value of our models.

An aspect that is important but not described well in the manuscript is the required model input compared to that of VGPM, CbPM, and CAFE. In one statement, the authors write: "These inputs overlap substantially with those used in VGPM, CbPM, and CAFE, demonstrating that the NN and Bayesian models do not require additional or more complex inputs." (l. 315). Later the manuscript states: "These probabilistic models do not require additional input variables beyond those used by VGPM, CbPM, and CAFE." (l. 720) Are really all 11 inputs listed in Table 1 used in VGPM, CbPM, and CAFE? Did the authors perform any experiments limiting the inputs to the NN and Bayes model further to examine which inputs are actually required to produce the output?

We appreciate the reviewer's meticulous examination of the input variable descriptions. We acknowledge that the original manuscript contained imprecise statements regarding input variables. Specifically, not all 11 variables listed in Table 1 were used by VGPM, CbPM, and CAFE; for instance, variables such as height of tide (m) and 1/10th significant wave height (m) are novel to our modeling framework. We have revised the relevant text (lines 320-321, 721) to clarify that our NN and Bayesian models extend beyond the input requirements of VGPM, CbPM, and CAFE, rather than matching them exactly. To assess input necessity, we conducted ablation experiments systematically removing individual input variables. In all cases, performance declined, confirming that each variable contributes meaningfully to model accuracy. Therefore, the full set of 11 variables was retained to ensure robust predictions. We hope this addresses the reviewer's concerns and provides sufficient context for the methodological choices made.

When the data used for training a NN or model is very limited, a common thing to do

is bootstrapping, i.e. dividing the data into different training and testing datasets repeatedly. Did the authors try different testing and training data configurations? It may shed more light on the differences in the CDF curves that are discussed in Section 3.2.2.

When dividing the training set and the dataset, different ratios have been tried to explore the model effectiveness in different cases, and the final chosen ratio is 8:2. Not only because the model evaluation metrics are better in this case, but also previous studies have indicated that the 8:2 ratio is a widely adopted practice in the field of machine learning and deep learning, which strikes a balance between providing a sufficiently large training set to efficiently learn features and patterns, and providing a smaller test set to robustly evaluate the generalization ability of the model. It strikes a balance between providing a sufficiently large training set to effectively learn features and patterns, and providing a smaller test set to robustly evaluate the generalization ability of the model.

**specific comments**

L 54: "Conventional methods of NPP measurement, such as ship-based sampling and bottle incubations, are beset with challenges like human errors and inadequacies in capturing spatial and temporal dynamics. This underscores the necessity for more sophisticated and comprehensive methods (Yang et al., 2021; Li et al., 2020)." True, but this study relies very much on monitoring data from a station and thus does not capture spatial dynamics -- it further relies on continuous measurements to capture the temporal dynamics. The authors mention this later: "Due to factors such as equipment malfunctions and adverse weather conditions, some data for the eleven variables were incomplete." (l. 198).

We agree that spatial variability remains a limitation of our current setup and have revised the manuscript (lines 55-60) to more clearly distinguish between spatial and temporal dynamics, and to acknowledge that our approach primarily addresses the latter. We also clarified the limitations posed by data gaps due to equipment malfunctions and environmental constraints (lines 201-203). We thank the reviewer for pointing out this important distinction.

L 79: "Currently, the most widely utilized models for estimating NPP include the

Vertically Generalized Production Model (VGPM), [...], have been proposed.": This sentence needs to be rephrased.

We have rephrased this sentence as "Currently, the estimation of NPP primarily relies on three mainstream models: the Vertically Generalized Production Model (VGPM), the Carbon-based Productivity Model (CbPM), and the Carbon, Absorption, and Fluorescence Euphotic-resolving Model (CAFE). These models were successively proposed by Behrenfeld et al. (1997), Westberry et al. (2008), and Silsbe et al. (2016), respectively, and have become benchmark methods in this research field." in line 81-86.

L 156: "The proportion of excellent water quality in Guangxi's near-shore waters reaches more than 90% all year round": It is not clear what this means. What is this measure of water quality, and is this based on a study or survey that could be cited? Similarly, what does "the quality of the marine ecological environment has remained at the forefront of the country" imply? More specific language and references would be useful here.

The description of the study area has been modified to "The island extends in a NE-SW direction and has an elliptical shape. It is approximately 6 km long from north to south, 5 km wide from east to west, and has an area of approximately 25 km2, making it the largest and youngest volcanic island in China (Li and Wang, 2004). Weizhou Island is an inhabited volcanic island, the annual average water surface temperature is about 24°C, and ranges from 19°C to 30°C. The annual average seawater salinity is 32‰, seawater pH ranges from 8.0 to 8.23, and seawater transparency ranges from 3 m to 10 m (Yu et al., 2019). In addition, Weizhou Island is the northernmost island in the Gulf of Tonkin, where coral reefs have developed. These coral reefs are mainly found in shallow waters along the southwest, northwest, and northeast coasts, with widths ranging from 0.86 to 2.56 km (He and Huang, 2019). " in line 159-169.

L 163: "Weizhou Island, located in the southern subtropical monsoon zone,

experiences a pleasant climate with abundant heat and precipitation throughout the year." Phrases like "pleasant climate" or "abundant heat and precipitation" are not specific or quantitative. The next sentence already specifies average (air?) temperatures, so the "pleasant climate" is not necessary here.

We have removed the adjectives like "pleasant" and "abundant" in the revised manuscript.

Eq. 1: Mention right away what theta and D represent in the equation.

An explanation of theta and D has been added to the text "where $\theta$ denotes the model parameters, and $D$ represents the training dataset" in line 277.

L 367: "In probabilistic forecasting, the focus extends beyond mere point estimates to encompass the shape and dispersion of the probability distribution.": This sentence and the next could go to the beginning of the section to give a better motivation for the use of CRPS.

We have repositioned these two sentences to the beginning of section 2.3.2.

L 382: "y the predicted value, x the observed value". This works, but is not conventional. Typically, x are the predicted values and y denotes observations.

We have modified the formula accordingly.

L 393: The CDF is introduced here, but it has already been used above in the definition of CRPS. I would suggest switching the section order.

The order of presentation of CDF and CRPS has been adjusted.

L 483: "On using CAFE as a prediction target, both models show more consistent performance.": The term model has now been used to describe VGPM, CbPM, and CAFE, but also the NN and Bayesian model. Please ensure that the reader always knows what models are referenced in the text. Furthermore, this statement about consistent performance for both models seems to contradict a later one: "In addition, for NN model's MAPD index value for CAFE is lower than that for Bayes model" (l

487).

A clearer representation of the individual models has been made in the text to avoid ambiguity. What this section is trying to convey is that both NN and Bayesian models have better performance when CAFE is used as the prediction target than when the other two NPPs are used as the prediction target, and the presentation in section 3.1 of the article has been adjusted.

L 490: "Overall evaluation indicates that under both models' assessment criteria, CAFE demonstrates superior accuracy in predicting effects compared to VGPM and CbPM.": This paragraph is not very helpful. What are the two assessment criteria used here? (Fig. 5 uses three metrics, not two.) What does "predicting effects" mean? It is not helpful to the reader that the remaining paragraph discuss VGPM and CbPM results and not CAFE.

Section 3.1 has been restructured to use three evaluation criteria, CRPS, RMSD and MAPD, which are presented in L454-456, and the lower the three metrics, the better the model performance. L460-487 analyze these three metrics for NN and Bayesian models when different NPPs are used as prediction targets in order to evaluate the performance, which reveals that CAFE is used as the prediction target. NN and Bayesian performance was more favorable when the target was CAFE, and thus CAFE was chosen as the main prediction target for subsequent analysis.

L 499: "(1) prior research indicating that CAFE provides relatively accurate estimates of NPP in marine ecosystems with characteristics similar to the Weizhou Island area, due to its advanced parameterization of phytoplankton dynamics". Please cite this prior research or provide some evidence for this statement.

We have added the reference as below.

"(1) Previous studies have shown that for other NPP models analyzed for the same dataset, the CAFE model explains the most variance and has the lowest model bias, and also reproduces the magnitude and seasonality of field-measured NPP better than other satellite remote sensing models (Silsbe et al., 2016)."

L 520: Is this analysis based on the testing data or the full CAFE-based dataset?

This analysis was based on the full CAFE-based dataset.

L 523: Are these confidence intervals credible intervals for the Bayesian model?

This is the confidence interval, which has been described in line 526-528.

L 590: "Fig. 8 demonstrates the CDF curves of the predicted mean values after the normalization process and the CDF curves of the CAFE." This sentence and the next are difficult to understand. Are they meant to emphasize the advantages of normalizing the values? Why make this point right after stating that divergence between these two CDFs should be minimal? Please rephrase.

We have rephrased this paragraph as below.

"Fig. 8 demonstrates the CDF curves of the predicted mean values after the normalization process and the CDF curves of the CAFE. The CDF plots of the normalized data can reflect the statistical distribution of the datasets, especially when the different datasets have different magnitudes or scales, and the normalization can eliminate these differences, which makes the comparisons and analyses between the different datasets more accurate and intuitive. Fig. 9 specifically quantifies the difference between the two CDF curves in Fig. 8 at each point, which is accomplished by calculating the difference between the y-values of the two CDF curves at the same x-value. Optimally, the divergence between these two CDFs should be minimal, manifested as extensive overlap between the yellow and blue curves in Fig. 8, and the blue curve in Fig. 9 approaching zero."

L 671: Is the only difference between the estimates in this section and previous ones the daily resolution?

We thank the reviewer for the question. Yes, the primary difference in this section is the temporal resolution. While earlier sections focused on 8-day estimates aligned with remote sensing data, this section presents daily NPP estimates derived from our models. The objective is to bridge gaps between remote sensing observations and enable finer-resolution analysis of NPP dynamics. This higher temporal resolution

also facilitates time series analysis to identify periodic patterns that may be obscured at coarser scales. We have clarified this point in the revised manuscript (line 677-683).

L 722: "By prioritizing variables such as SST and AP, the models can be optimized to reduce reliance on less influential inputs, improving efficiency without compromising accuracy." Was this actually shown? Did the authors try to run the NN or Bayes model with fewer input variables?

To assess input necessity, we conducted ablation experiments systematically removing individual input variables. In all cases, performance declined, confirming that each variable contributes meaningfully to model accuracy. Therefore, the full set of 11 variables was retained to ensure robust predictions.